# Capture of single Ag atoms through high-temperature-induced crystal plane reconstruction

Jiaxin Li[1,6], Kai Li[1,2,6], Zhao Li [ID][1], Chunxue Wang[1], Yifei Liang[1], Yatong Pang[1], Jinzhu Ma [ID][3,4,5] ✉, Fei Wang [ID][1,2] ✉, Ping Ning[1,2] & Hong He [ID][3,4,5]

The "terminal hydroxyl group anchoring mechanism" has been studied on metal oxides ($Al_2O_3$, $CeO_2$) as well as a variety of noble and transition metals (Ag, Pt, Pd, Cu, Ni, Fe, Mn, and Co) in a number of generalized studies, but there is still a gap in how to regulate the content of terminal hydroxyl groups to influence the dispersion of the active species and thus to achieve optimal catalytic performance. Herein, we utilized AlOOH as a precursor for γ-$Al_2O_3$ and induced the transformation of the exposed crystal face of γ-$Al_2O_3$ from (110) to (100) by controlling the calcination temperature to generate more terminal hydroxyl groups to anchor Ag species. Experimental results combined with AIMD and DFT show that temperature can drive the atomic rearrangement on the (110) crystal face, thereby forming a structure similar to the atomic arrangement of the (100) crystal face. This resulted in the formation of more terminal hydroxyl groups during the high-temperature calcination of the support (Al-900), which can capture Ag species to form single-atom dispersions, and ultimately develop a stable and efficient single-atom Ag-based catalyst.

In heterogeneous catalysis, a strong metal-support interaction (SMSI) is deemed to be a key factor affecting the anchoring of metals on the surface of reducible oxides and plays a significant role in influencing the dispersion of the active component[1,2], the electron migration between the metal and the support, and thus the performance of the catalytic reaction. However, there are almost no reports regarding SMSI on non-reducible oxide supports. Nevertheless, catalysts supported on non-reducible oxides are also widely applied. For non-reducible oxides like $Al_2O_3$, which serves as a support for the widely used $Ag/Al_2O_3$ catalysts, typical reactions include selective catalytic oxidation of $NH_3$ ($NH_3$-SCO)[3,4], selective catalytic reduction (SCR) of $NO_x$ with $NH_3$ ($NH_3$-SCR)[5–7], CO oxidation[8,9], (hydrocarbon-based selective catalytic reduction of $NO_x$) HC-SCR[10–12], ethylene epoxidation reaction[13,14], etc. Therefore, it is necessary to clarify the anchoring mechanism of Ag on the $Al_2O_3$ surface to regulate the state of Ag active species and thus enhance the catalytic reaction performance.

In our previous study[15–17], we used two types of γ-$Al_2O_3$, with different hydroxyl contents, nano-sized γ-$Al_2O_3$ and micro-sized γ-$Al_2O_3$, and found that Ag species are mainly anchored to γ-$Al_2O_3$ through terminal hydroxyl group. DFT calculations revealed that the (100) surfaces of γ-$Al_2O_3$ have more terminal hydroxyl groups than the (110) surfaces, allowing Ag to be anchored as a single atom by the terminal hydroxyls. When not enough terminal hydroxyl groups are available to anchor the Ag species, they tend to aggregate into Ag clusters and Ag NPs[15]. Based on the proposed "terminal hydroxyl group anchoring mechanism," we constructed more effective $NH_3$-SCO active sites

[1]Faculty of Environmental Science and Engineering, Kunming University of Science and Technology, 650500 Kunming, China. [2]National-Regional Engineering Center for Recovery of Waste Gases from Metallurgical and Chemical Industries, 650500 Kunming, China. [3]State Key Joint Laboratory of Environment Simulation and Pollution Control, Research Center for Eco-Environmental Sciences, Chinese Academy of Sciences, 100085 Beijing, China. [4]Center for Excellence in Regional Atmospheric Environment, Institute of Urban Environment, Chinese Academy of Sciences, 361021 Xiamen, China. [5]University of Chinese Academy of Sciences, 100049 Beijing, China. [6]These authors contributed equally: Jiaxin Li, Kai Li. ✉e-mail: jzma@rcees.ac.cn; wangfei@kust.edu.cn

using the "pre-occupied anchoring-site" strategy[16]. We used Cu with stronger anchoring strength to pre-occupy the anchoring sites to force Ag agglomeration, resulting in the construction of an efficient NH₃-SCO catalyst at low Ag loading. Meanwhile, we found that the "terminal hydroxyl group anchoring mechanism" is also applicable to the anchoring of other non-precious metals (Fe, Co, Ni, and Mn) on $Al_2O_3$ support. In addition, we found that on the $CeO_2$ support, the terminal hydroxyl group is also an anchoring site for Ag atoms, and terminal OH group on the $CeO_2$ (100) surface can firmly anchor Ag via the formation of a dumbbell structure[17]. Moreover, other metals (Pt, Pd, etc.) on $CeO_2$ can also be directly anchored to terminal hydroxyl group.

As mentioned above, the above findings suggest that terminal hydroxyl groups on oxide surfaces are the anchoring sites for metals, and that the type of exposed crystal facets ($Al_2O_3$ (100), $CeO_2$ (100)) on the metal oxide significantly affects the content of the terminal hydroxyl groups, and thus the dispersion of the metal. However, it is not clear how to regulate the content of terminal hydroxyl groups on the support surface or the type of exposed crystal faces and whether the transformation of the crystal plane type correspondingly affects the type and content of hydroxyl groups. This is thus clearly worthy of further investigation. In conventional nanocatalysis, the influence of the crystal facet effect of supports on the activity of the catalytic reaction has been extensively studied. Hu et al.[18] loaded Pd on different crystal facets of $CeO_2$ and found that on the $CeO_2$ (100) facets, Pd exists predominantly in the form of Pd SAs (single atoms). In contrast, on the $CeO_2$ (111) facets, Pd readily aggregates into Pd clusters. Notably, when the size of the catalytically active components goes from the nanoscale to the single-atom scale, the crystal facet effect becomes more noticeable and must be considered.

Physically, when a crystal surface is exposed to different thermodynamic conditions, a change in surface energy occurs, which can lead to the possibility of atomic rearrangement. In this study, we aimed to investigate the effect of the type of exposed crystal surface of γ-$Al_2O_3$ on the surface hydroxyl content and Ag anchoring through high-temperature-induced crystal surface transformation. We used AlOOH as the precursor for γ-$Al_2O_3$ and calcined it at different temperatures (500 °C, 600 °C, 700 °C, 800 °C, 900 °C). Using a range of characterization techniques, we found that the lattice spacing of γ-$Al_2O_3$ changed during high-temperature calcination, resulting in the transition from (110) to (100) crystal facets, which exposed more terminal hydroxyl groups and allowed Ag to exist as single atoms on samples calcined at 900 °C. Additionally, the Ag/(Al-900) sample exhibited better activity in the HC-SCR reaction and $O_3$ decomposition than the Ag/(Al-500) sample.

## Results

### Catalyst crystal transition process

The XRD pattern of fresh AlOOH and in situ XRD were used to demonstrate the crystal transition process of AlOOH calcined at different temperatures, as shown in Supplementary Fig. 1, 2. The AlOOH crystalline phase persists at calcination temperatures up to 300 °C and starts taking on the crystalline form of γ-$Al_2O_3$ at 400 °C. Diffraction peaks at 37.5°, 45.7°, 60.5°, 66.6°, and 84.5° were observed, corresponding to the γ-$Al_2O_3$ (311), (400), (511), (440), and (444) crystal planes (JCPDS 02-1420), which indicates that AlOOH is converted to γ-$Al_2O_3$ starting at about 400 °C. Furthermore, these peaks increased in intensity with higher calcination temperatures, suggesting that the crystallinity of γ-$Al_2O_3$ gradually improved with increasing calcination temperature. Notably, there were no diffraction peaks other than γ-$Al_2O_3$ in the temperature range of 500–900 °C, suggesting that no crystalline phase transformation occurred at 900 °C, and AlOOH was completely converted to γ-$Al_2O_3$.

The $N_2$ adsorption and desorption isotherms of the Ag-loaded AlOOH catalysts were also analyzed, as well as the pore size distribution (Supplementary Fig. 3). The adsorption and desorption isotherms

for the catalysts belonged to type III and were accompanied by an H3-type hysteresis loop[19]. This indicated that a significant number of pores with sizes ranging from 1 to 100 nm were formed due to the loose binding of $Al_2O_3$ particles in the catalyst. The pore structure of the catalyst did not undergo significant changes under different calcination temperatures. The specific surface area, average pore size, and pore volume of the catalyst were also measured (Supplementary Table 1), showing that as the calcination temperature increased, the average pore size of the catalyst gradually increased, leading to a decrease in specific surface area and pore volume. This was attributed to the collapse and consolidation of some micropores in the support during high-temperature calcination. However, nanoscale pores in the range of 3–8 nm remained the dominant pore size, and the overall pore structure did not undergo significant changes.

To investigate the crystalline transformation process of AlOOH calcined at different temperatures, HR-TEM was used to characterize the types of crystal surfaces exposed on the surface of γ-$Al_2O_3$ obtained after calcination. The lattice spacing of γ-$Al_2O_3$ changed notably at different calcination temperatures, indicating changes in the exposed surfaces (Fig. 1). For instance, at 500 °C, the lattice spacing corresponded to the γ-$Al_2O_3$ (440) crystal plane, belonging to the (110) crystal plane group. As the calcination temperature increased, the lattice spacing gradually increased. At 900 °C, the lattice spacing corresponded to the γ-$Al_2O_3$ (400) crystal plane, which belongs to the (100) crystal plane group[20]. This suggests that high-temperature calcination transforms the exposed crystal surface of γ-$Al_2O_3$ from the (110) crystal surface group to the (100) crystal surface group. Furthermore, ab initio molecular dynamics simulation (AIMD) was employed to confirm the transformation of γ-$Al_2O_3$ crystal plane at high temperatures. The γ-$Al_2O_3$ (110) surface was initially designed (Fig. 1f) and fully relaxed at 1173 K for 40 ps. The results showed that the γ-$Al_2O_3$ (110) surface underwent significant structural changes at this high temperature (Fig. 1g, h). Atoms on the surface rearranged to form an atomic arrangement similar to that on the γ-$Al_2O_3$ (100) crystal surface (red arrows), indicating a tendency for the (110) surface to transform into the (100) surface under high-temperature conditions. Additionally, surface free energy calculations using density functional theory (DFT) revealed that the surface free energy of γ-$Al_2O_3$ (110) increased with rising calcination temperature, while that of γ-$Al_2O_3$ (100) decreased with increasing temperature (Fig. 1i). The surface energy of γ-$Al_2O_3$ (100) at 1173 K was significantly lower than that of γ-$Al_2O_3$ (110), further supporting the transformation of the (110) surface to the (100) surface at elevated temperatures.

### Relationship between hydroxyl content and calcination temperature

Building upon our previous findings, DFT theoretical calculations revealed that the γ-$Al_2O_3$ (100) crystal plane possesses more terminal hydroxyl groups compared to the γ-$Al_2O_3$ (110) crystal plane[15]. Consequently, our study delved deeper into the evolution of surface hydroxylation on AlOOH at varying temperatures, employing in situ DRIFTS spectra of NH₃ adsorption to analyze the surface hydroxyl groups of AlOOH calcined at different temperatures. As depicted in Fig. 2a, three negative peaks were identified at 3791 cm⁻¹, 3740 cm⁻¹, and 3678 cm⁻¹, corresponding to type I, II, and III hydroxyls, respectively[15,21–24]. The consumption peaks for the three types of hydroxyl groups signify the presence of diverse surface hydroxyl groups on the γ-$Al_2O_3$ surface obtained through calcination of AlOOH at varying temperatures. Furthermore, in situ DRIFTS of NH₃ adsorption demonstrated a notable shift in the type of hydroxyl groups on γ-$Al_2O_3$ as the calcination temperature increased, with a significant enhancement in the intensity of terminal hydroxyl groups (type I)[21,25]. In addition, in order to further clarify the physical effects of high-temperature calcination on the formation of hydroxyl groups on AlOOH, we also examined the hydroxyl group changes under different atmospheres and calcination times

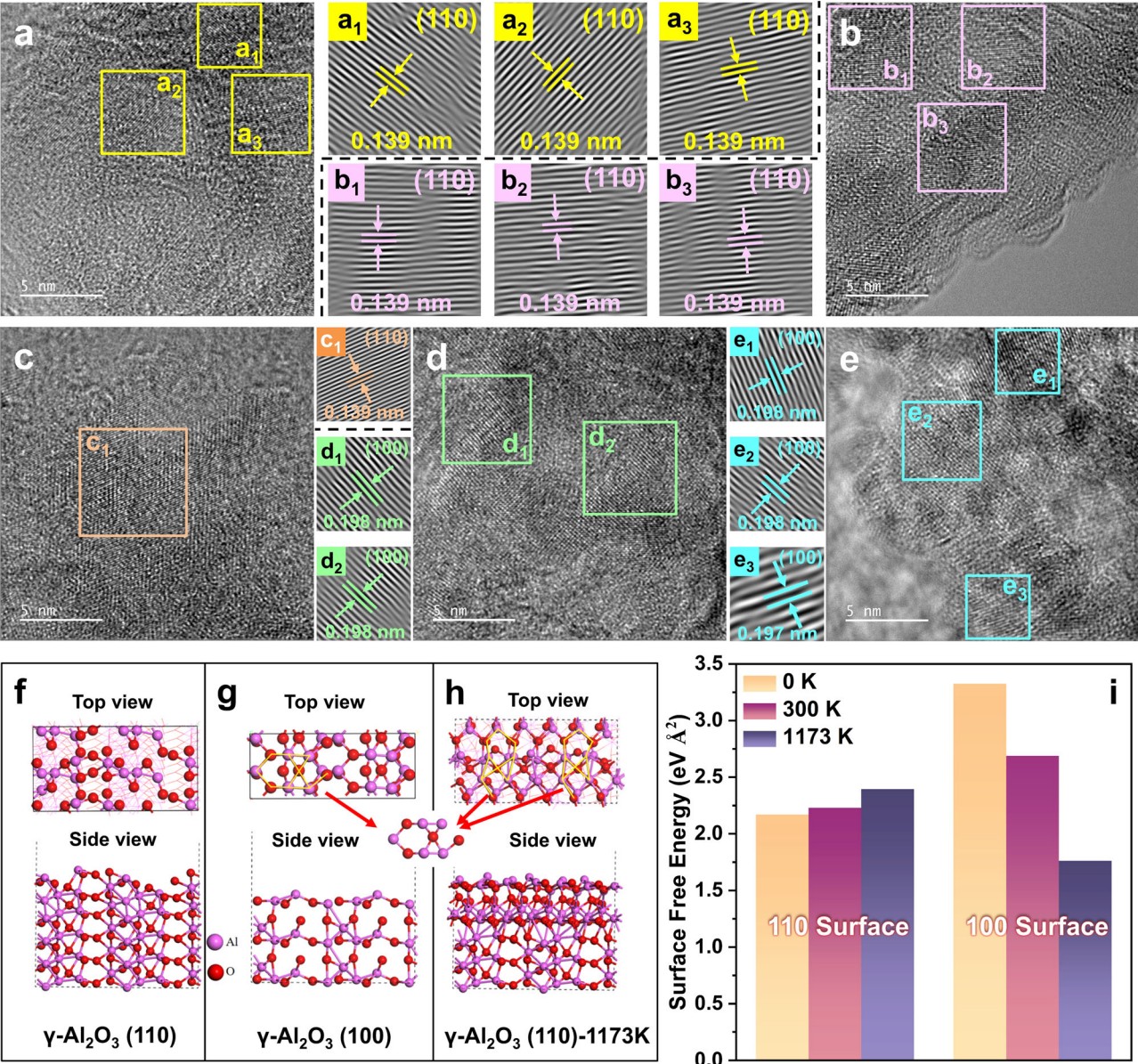

**Fig. 1 | The crystal plane transformation process of AlOOH calcined at different temperatures, and AIMD and DFT calculations. a–e** HR-TEM images of Al-500 °C, 600 °C, 700 °C, 800 °C, 900 °C (a1-e3 are the Fast Fourier transforms (FFT) of the respective color regions). **f** γ-Al$_2$O$_3$ (110) and (**g**) γ-Al$_2$O$_3$ (100) crystal surface structures. **h** γ-Al$_2$O$_3$ (110) crystal surface structure at 1173 K. **i** Surface free energy of γ-Al$_2$O$_3$ (110) and γ-Al$_2$O$_3$ (100) structures at different temperatures (0 K, 300 K, and 1173 K).

(Supplementary Fig. 4a–c). Overall, changing the calcination atmosphere and time changed the hydroxyl content to varying degrees, and after screening the conditions, the optimal terminal hydroxyl and total hydroxyl contents were achieved for the 500 °C and 900 °C samples by calcining in an air atmosphere for 3h (Supplementary Fig. 4d), and this calcination condition was also used in this work. Subsequently, we conducted in situ DRIFTS spectroscopy of NH$_3$ adsorption on the 1% Ag-loaded sample, as shown in Fig. 2a (gray line). This analysis indicates that the surface hydroxyl content of γ-Al$_2$O$_3$ obtained by calcination of AlOOH at different temperatures tends to decrease with the introduction of Ag. The most pronounced changes were observed in the terminal hydroxyl group peak at 3791 cm$^{-1}$. Thus, this further corroborates our earlier findings that terminal hydroxyl groups are the primary anchoring sites for Ag. The concentration of surface OH groups was then quantified by fitting distinct peaks for samples before and after Ag loading at different calcination temperatures (Fig. 2b–d). Next, we established linear correlations between the calcination temperature and the peak area of

different hydroxyl types by integrating these peak areas. As demonstrated in Supplementary Fig. 5, the peak area of terminal hydroxyl groups (type I) progressively increasesd with rising calcination temperature, displaying a strong positive correlation, with an R$^2$ value of 0.95. In contrast, the other two hydroxyl groups showed no significant association with calcination temperature. Specifically, the terminal hydroxyl content at the beginning of the loading process was 3.6%, 7.7%, 8.7%, 11.1%, and 12.4%, respectively, for samples calcined at temperatures ranging from 500 °C to 900 °C. Upon Ag loading, the terminal hydroxyl content at the end of loading significantly decreasesd to 2.3%, 3.3%, 4.4%, 5.8%, and 6.3% (refer to Supplementary Table 2 for detailed information). These results strongly suggest that the terminal hydroxyl content undergoes a dramatic increase with increasing calcination temperature. The substantial reduction in terminal hydroxyl content following Ag loading reinforces the notion that Ag primarily anchors to γ-Al$_2$O$_3$ through terminal hydroxyl groups (Fig. 2e). The augmentation of terminal hydroxyl group content through high-temperature calcination

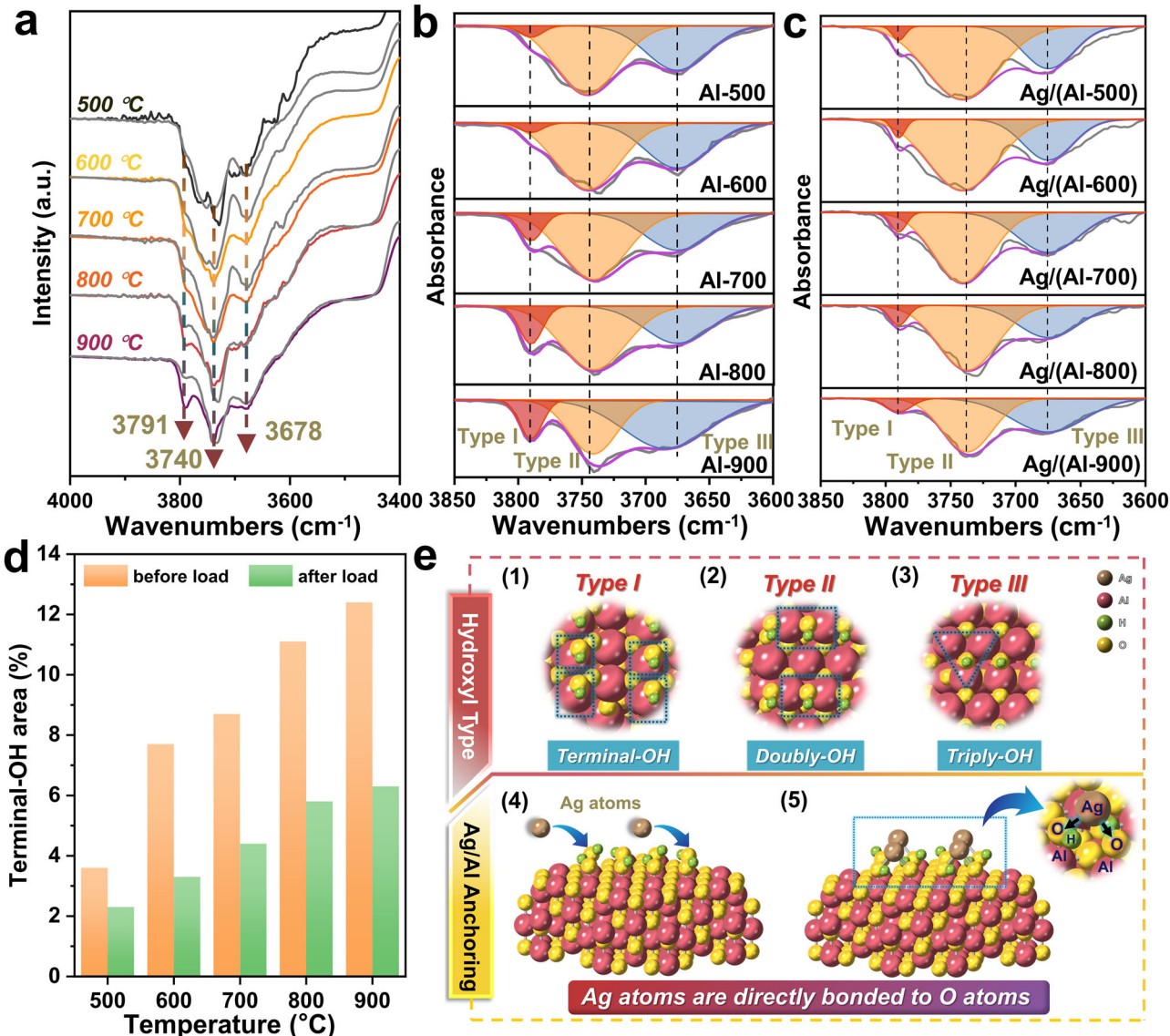

**Fig. 2 | Hydroxyl change characterization and Ag anchoring mechanism schematic diagram. a** Trends in hydroxyl groups before and after Ag loading at different calcination temperatures (Ag loadings (0, 1%)) (the colored line represent before Ag loading, and the gray line represent after Ag loading.). **b** Resolution of OH consumption peaks after in situ DRIFTS of $NH_3$ adsorption over AlOOH at different temperatures (shaded regions (red, yellow, and blue) represent the Type I, Type II, and Type III hydroxyl groups, respectively). **c** Resolution of OH consumption peaks after in situ DRIFTS of $NH_3$ adsorption over Ag/(Al-500 °C, 600 °C, 700 °C, 800 °C, 900 °C) at different temperatures. **d** Terminal OH contents before and after Ag loading determined through in situ DRIFTS measurements at different calcination temperatures. **e** Schematic illustration of the Ag anchoring mechanism (e1-e3 blue shapes represent the difference in coordination numbers between O and Al among the three types of hydroxyl groups, e4 blue arrows represent the process of Ag anchoring on the surface of γ-$Al_2O_3$, and e5 blue arrows represent the detailed diagram of Ag anchored on γ-$Al_2O_3$ through terminal hydroxyl groups).

establishes the groundwork for regulating the dispersion of Ag particles on γ-$Al_2O_3$.

## Dispersion of Ag species

X-ray diffraction (XRD) patterns of Ag/(Al-X) samples with different calcination temperatures showed no diffraction peaks associated with Ag species, indicating that Ag was well dispersed on the γ-$Al_2O_3$ support and did not agglomerate significantly as the calcination temperature increased (Fig. 3a). HAADF-STEM images of the samples revealed that Ag clusters were observed on the Ag/(Al-X) (X = 500, 600, 700 and 800 °C) samples (Fig. 3b−e red circles), while the Ag/(Al-900) sample exhibited single-atom dispersion of Ag (Fig. 3f red circles). This dispersion behavior was attributed to the presence of a large number of terminal hydroxyl groups on Ag/(Al-900), which provided ample anchoring sites for Ag, allowing for atomic dispersion. ICP

results indicated that the Ag content for both the Ag/(Al-500) and Ag/(Al-900) samples was 0.91 wt% (Supplementary Fig. 6).

In addition to the fact that the number of terminal hydroxyl groups plays an important role in the dispersion of Ag species, it is unclear whether the uniform distribution of hydroxyl groups also has an effect on Ag species. Therefore, we constructed models with unevenly distributed terminal hydroxyl groups on the (100) crystal plane of γ-$Al_2O_3$ and uniformly distributed hydroxyl groups on the (110) crystal plane to verify this (Fig. 3g). After structural optimization, we found that on the (100) crystal plane, Ag remains in the form of single atoms on the surface of γ-$Al_2O_3$, while on the (110) crystal plane, Ag agglomerates into small clusters. This indicates that the uniformity of the hydroxyl group distribution is not the primary factor affecting Ag dispersion; rather, the number of terminal hydroxyl groups is the key factor determining Ag dispersion.

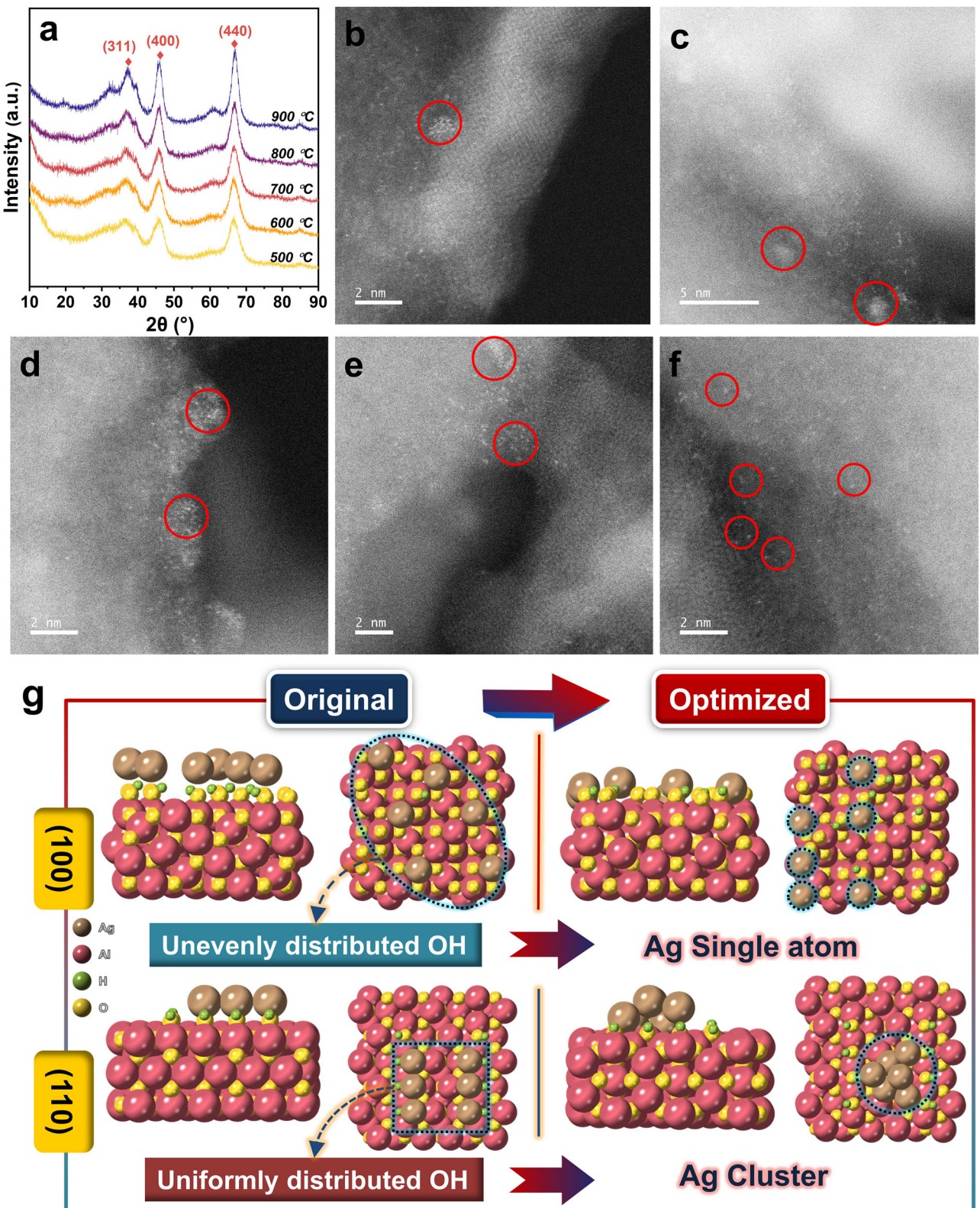

**Fig. 3 | Characterization of the dispersion of Ag species and DFT calculations. a** XRD profiles after loading Ag species. **b–f** HAADF-STEM images of Ag dispersion on Ag/(Al-500 °C, 600 °C, 700 °C, 800 °C, 900 °C) samples (red circles represent the change from Ag clusters to Ag single-atom dispersions). **g** Ag species on the γ-Al$_2$O$_3$ (100) and (110) crystal planes after structural optimization.

## Valence, coordination condition, and charge density difference of Ag species

Supplementary Fig. 7 shows XPS data for O, Al, and Ag. In the O 1$s$ spectrum, peaks located at around 531.1 eV and 532.2 eV were detected, respectively, with the former attributed to lattice oxygen (O$_L$) and the latter value assigned to surface OH (O$_{OH}$) species[26,27]. Clearly, we can see that the proportion of O$_{OH}$ decreases from 41% in the Ag/(Al-500) sample to 30% in the Ag/(Al-900) sample, indicating

that the total OH content decreases as the calcination temperature of AlOOH is increased, which is also consistent with our in situ DRIFTS results. In the Al 2$p$ spectrum, the peaks for the Ag/(Al-500) and Ag/(Al-900) samples are located at 74.3 eV and 74.6 eV, respectively, which correspond to the 2$p$ peaks of the oxidized state of $Al_2O_3$[28], confirming that the AlOOH was indeed converted to $Al_2O_3$ after calcination. We can observe that for the XPS spectra at the Ag 3$d$ core level, the intensity of the Ag/(Al-900) sample is stronger compared to the Ag/(Al-500) sample, indicating that the large number of terminal hydroxyl groups on the surface of the sample induces greater Ag dispersion, resulting in an increase in the number of Ag atoms on the surface in the region. Secondly, the change in calcination temperature also caused a change in the valence state of Ag. The Ag 3$d_{5/2}$ peak of the Ag/(Al-500) sample is located at 368.2 eV, which is close to the position of the

metallic silver energy peak. The binding energy of Ag in the Ag/(Al-900) sample shifted to 368.7 eV, suggesting that the Ag species in the Ag/(Al-900) sample existed mainly in the oxidized state (Ag$^+$)[29,30].

The Ag-K edge XAFS data for Ag/(Al-500), Ag/(Al-900), Ag foil, AgNO$_3$, and Ag$_2$O are presented in Fig. 4. From the normalized near-edge structure (XANES) (Fig. 4a), the intensity of the white line peak is located between those of AgNO$_3$ and Ag foil, which means that the valence state of Ag species lies between 0 and 1[31]. Specifically, the Ag/(Al-500) sample exhibits a preference for the Ag foil's valence state, whereas the valence of Ag in the Ag/(Al-900) sample shifts toward that of AgNO$_3$ with increasing calcination temperature. The Fourier transforms of the $k^2$-weighted EXAFS spectra are displayed in Fig. 4b. As shown in the figure, there are two distinct peaks at 1.65 Å and 2.7 Å attributed to Ag–O and metallic Ag–Ag bonding, respectively[32–35].

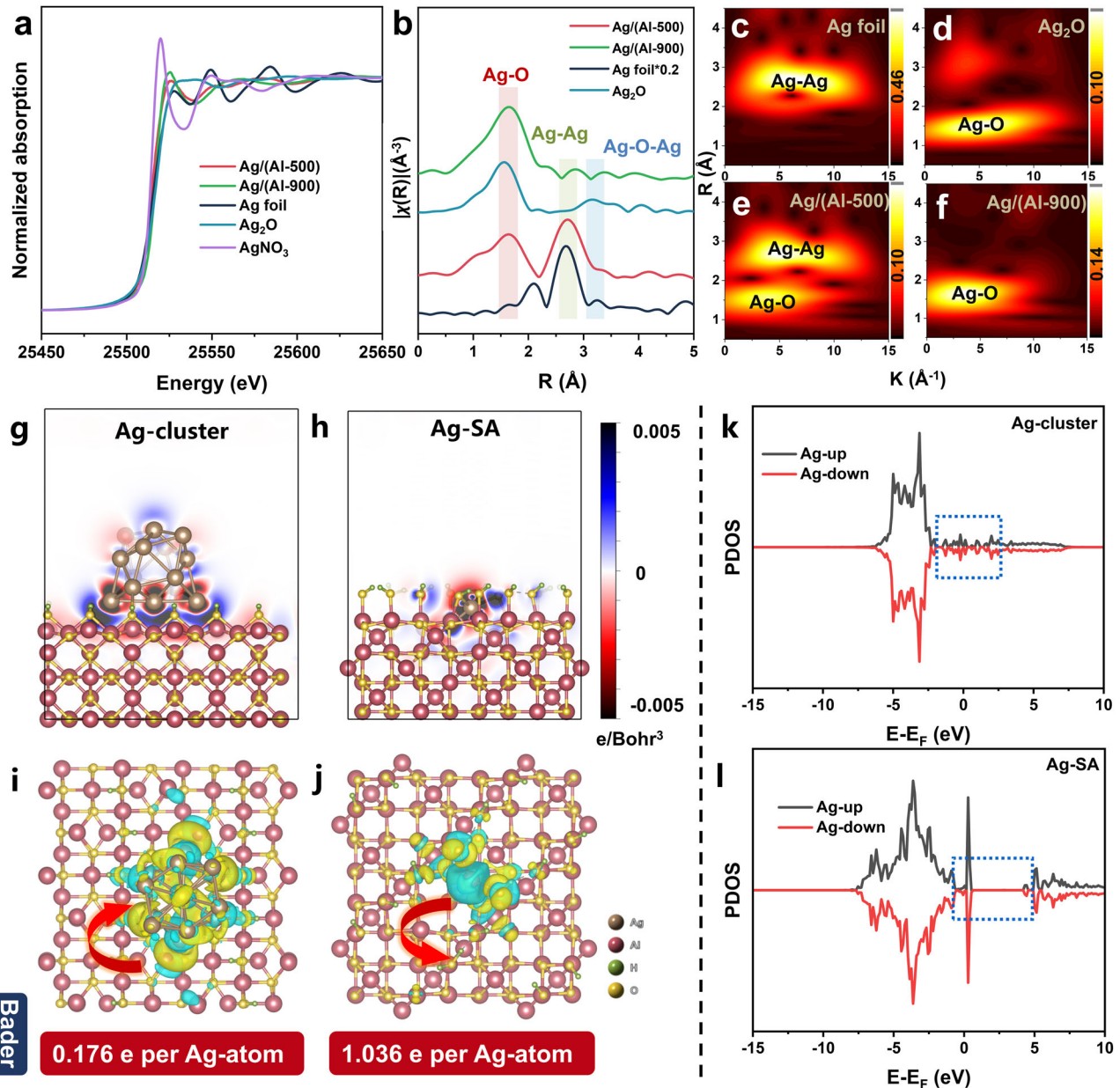

**Fig. 4 | Characterization of valence, coordination condition and charge density difference of Ag species. a** Ag-K edge XANES spectra of standard samples and Ag/(Al-500), Ag/(Al-900). **b** Fourier transform of Ag K-edge EXAFS spectra (shaded regions (red, green, and blue) represent Ag–O, Ag–Ag, and Ag–O–Ag coordination peaks, respectively). **c–f** WT of Ag foil, Ag$_2$O, Ag/(Al-500) and Ag/(Al-900). **g–j** Charge density difference analysis of Ag/(Al-500) (**g, i**) and Ag/(Al-900) (**h, j**). Cyan and yellow regions indicate electron depletion and accumulation, respectively, the red arrow represents the electron transfer between the γ-Al$_2$O$_3$ and Ag, and the corresponding Bader charge is shown at the bottom of the charge density map. **k–l** PDOS of Ag/(Al-500) (**k**) and Ag/(Al-900) (**l**).

Compared to the spectra of Ag foil and $Ag_2O$, the Ag/(Al-500) sample exhibits coordination peaks at Ag–O and Ag–Ag with coordination numbers of 2.9 and 3.6, respectively (see Supplementary Table 3, and Supplementary Fig. 8, 9 for fitting parameters and curves). In contrast, the Ag/(Al-900) sample shows only a peak at 1.65 Å with an Ag–O coordination number of 3.6, and no Ag–Ag coordination peak is observed at 2.7 Å, unlike the Ag/(Al-500) sample. For a more precise description of the Ag dispersion state and coordination conditions, wavelet transform (WT)-EXAFS was performed due to its high resolution in both k and R space[36–41] (Fig. 4c–f). The WT contour plots for Ag/(Al-500) show two signals centered at 4.0 Å$^{-1}$ and 5.8 Å$^{-1}$, corresponding to Ag–O and Ag–Ag contributions. In contrast, no signal corresponding to Ag–Ag coordination is observed in the plot for Ag/(Al-900). Only a strong Ag–O signal at approximately 4.0 Å$^{-1}$ is evident. This result firmly excludes the possibility of the existence of Ag clusters under 900 °C calcination conditions, further confirming that Ag is present as single atoms on the Ag/(Al-900) sample, while Ag clusters are observed on the Ag/(Al-500) sample.

Then we further investigated the charge density difference of Ag species in the Ag/(Al-500) and Ag/(Al-900) catalysts. According to the charge density difference and Bader charge data (Fig. 4g–j), it can be seen that the substrate transfers an average of 0.176 charge to each Ag atom in the cluster model (Ag/(Al-500)), whereas Ag transfers 1.036 charge to the substrate in the monatomic model (Ag/(Al-900)), and it is clear that the clusters are close to the metallic state, whereas in the monatomic system, Ag is stabilized by the surrounding OH, and the individual Ag atom apparently loses its charge and transfers electrons to O, exhibiting an oxidized state. Figure 4k, l show the density of states (DOS) for the cluster (Ag/(Al-500)) and Ag single atom (Ag/(Al-900)) models. In the cluster model, the d-band of Ag is continuous near the Fermi energy level and is metallic in nature (Fig. 4k blue box), while the d-band of monatomic Ag has a significant band gap near the Fermi energy level and has taken on semiconducting properties, indicating an oxidized state (Fig. 4l blue box).

## Performance test

To assess the versatility and applicability of the catalysts, we tested the catalytic activity of the Ag/(Al-500) and Ag/(Al-900) samples in different reactions. HC-SCR technology is considered to be a promising denitrification method, capable of simultaneously removing both $NO_x$ and hydrocarbons from flue gas[10,12]. Alumina-supported silver (Ag/$Al_2O_3$) catalysts find widespread use in HC-SCR applications[42,43]. Studies have indicated that highly dispersed silver cations (Ag$^+$) are the active centers in this reaction[11,15,44,45]. As depicted in Fig. 5a, Ag/(Al-X) (X = 500, 900) samples were employed to assess the $C_3H_6$-SCR reactivity, investigating the effects of changes in Ag species dispersion at different calcination temperatures. Notably, the Ag/(Al-900) samples exhibited higher activity at temperatures ranging from 150 to 400 °C compared to the Ag/(Al-500) samples. We hypothesize that samples calcined at 900 °C with single-atom dispersion are more active than those at 500 °C, which display Ag clusters. Thus, in conjunction with our previous research, it can be further deduced that Ag cations (Ag$^+$) exhibit enhanced reactivity in the hydrocarbon-selective catalytic reduction of $NO_x$ through HC-SCR. In addition, in the $O_3$ decomposition reaction (Fig. 5b), the Ag/(Al-900) sample maintained an ozone conversion of about 55% over 6 h, which was much greater than that of the Ag/(Al-500) sample, at about 15%.

A catalyst's stability is another essential indicator of its performance. For the Ag/(Al-900) sample, Fig. 5c displays $NO_x$ conversion of about 100% and 75% for 50 h at reaction temperatures of 350 °C and 300 °C, respectively. Subsequently, we further verified the stability of Ag/(Al-900) samples through ab initio molecular dynamics simulation. As shown in Supplementary Fig. 10, the system is stable both in terms of temperature and potential energy. We also calculated the Ag–O and Ag–Ag atomic distances (Fig. 5d), and the results showed that the average distance between Ag and the surrounding O atoms is always shorter than Ag–O bonds within $Ag_2O$ crystals. In addition, the distance between two Ag atoms is always longer than the Ag–Ag bond in the bulk, which indicates that the terminal hydroxyl groups on the γ-$Al_2O_3$ surface can stably anchor Ag single atoms without agglomeration. Therefore, during the simulation time of 10000 fs, Ag can be stably anchored on the γ-$Al_2O_3$ (100) surface at 773 K while maintaining the single-atom form (Fig. 5e), suggesting the good thermal stability of Ag/(Al-900).

## High-temperature capture process of single atoms

Based on the aforementioned results, we demonstrated the trapping of Ag single atoms by high-temperature calcination of AlOOH-induced crystal plane transitions to form more terminal hydroxyl groups. Next, we continued to investigate whether the terminal hydroxyl groups could form in situ and induce Ag dispersion in the presence of co-calcination of the substrate with Ag. We directly loaded Ag onto γ-$Al_2O_3$ (formed by AlOOH after 500 °C calcination), and there was a significant difference in color between the two samples after co-calcination of Ag and γ-$Al_2O_3$ at 500 and 900 °C, respectively. Noble metal nanoparticles (NPs) are known to exhibit distinct colors in the visible region due to light absorption and scattering caused by plasmon resonance[46,47]. This color variation is dependent on factors such as particle size, shape, refractive index of the surrounding medium, and particle spacing[48]. The color of the Ag/(Al-500)-500 sample appears yellow, while the color of the Ag/(Al-500)-900 sample turns white and the Ag particle size transforms from Ag clusters to highly dispersed single atoms (Fig. 6a–d). This result confirms that during the high-temperature co-calcination process, γ-$Al_2O_3$ can also achieve the transformation of crystal planes to construct a large number of terminal hydroxyl groups, providing sufficient anchoring points for Ag and facilitating the high dispersion of silver single atoms at elevated temperatures (Fig. 6e, f).

## Discussion

Single-atom catalysts gained favor among researchers due to their lower precious metal loadings, efficient atomic utilization, and well-defined active center[49–53]. For precise control of single atoms to optimize catalytic performance, various methods have been proposed. In addition to enhancing the strong metal-support interaction, trapping single atoms on defect-rich metal oxides[54,55], stabilization of single atoms by MOF structures[56], and specific strategies such as laser ablation[57]. In this work, we successfully prepared single-atom Ag based on the "terminal hydroxyl group anchoring mechanism" by high-temperature-induced crystal plane transition. During this process, Ag clusters gradually transform into individual Ag atoms, resulting in a more uniform dispersion of Ag species. This phenomenon is confirmed by the results obtained from ICP, XPS, and HAADF-STEM analyses. In both the Ag/(Al-500) and Ag/(Al-900) samples, the Ag content remains consistent. However, the number of Ag atoms on the surface of Ag/(Al-900) samples is notably higher.

In a nutshell, we introduced a straightforward method for obtaining Ag single atoms supported on γ-$Al_2O_3$ through high-temperature calcination-induced crystal facet changes, which can serve as a potential sintering-resistant catalyst. As the calcination temperature increases, the exposed crystal surface of γ-$Al_2O_3$ undergoes reconstruction, transitioning from (110) to (100), as indicated by HR-TEM and AIMD results. DFT calculations reveal that the (110) crystalline surface of γ-$Al_2O_3$ exhibits an inclination toward increased surface energy at high temperatures, while the (100) surface energy decreases with rising temperature, ultimately making the (100) surface energy lower than that of (110) at 1173 K. Hence, the (110) surface structure can potentially transition to (100) at elevated temperatures. This subsequent crystal surface reconstruction significantly alters the content and types of surface hydroxyl groups on γ-$Al_2O_3$. After calcination at 900 °C, more terminal

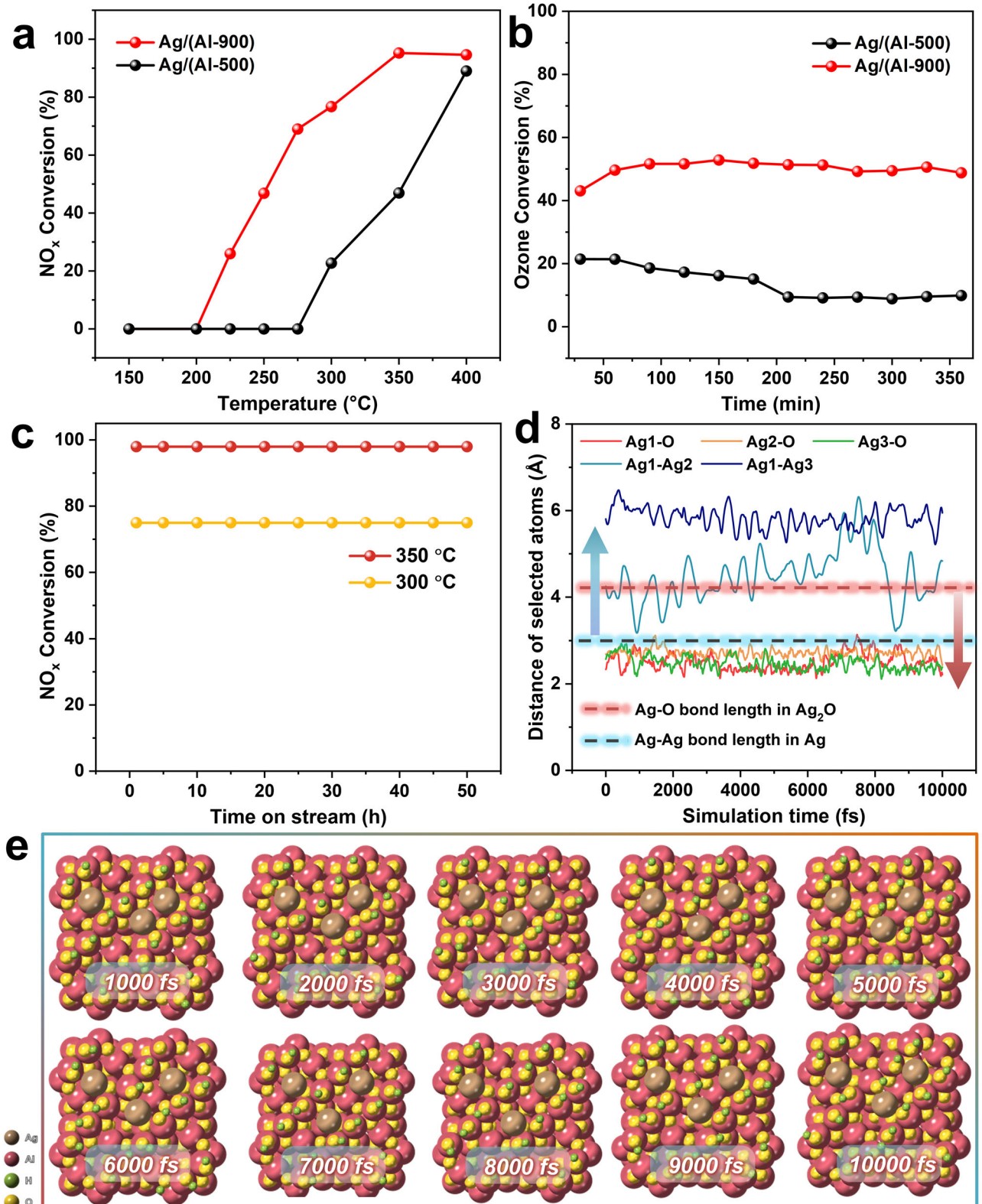

**Fig. 5 | Activity and stability tests. a** NO$_x$ conversion for C$_3$H$_6$-SCR (NO 800 ppm, C$_3$H$_6$ 1600 ppm, H$_2$ 1%, O$_2$ 10%, N$_2$ balance. GHSV 118,000 h$^{-1}$). **b** Ozone conversion for O$_3$ decomposition (SV = 840 L/g/h, dry, T = 30 °C, O$_3$ = 40 ppm). **c** Stability test for Ag/(Al-900) sample at 300 °C and 350 °C. **d** Time dependence of Ag1–O, Ag2–O, Ag3–O, Ag1–Ag2, and Ag1–Ag3 bond lengths (Å) during the ab initio molecular dynamics simulation (10000 fs). **e** Snapshot of single Ag atom on the γ-Al$_2$O$_3$ (100) surface at different simulation times.

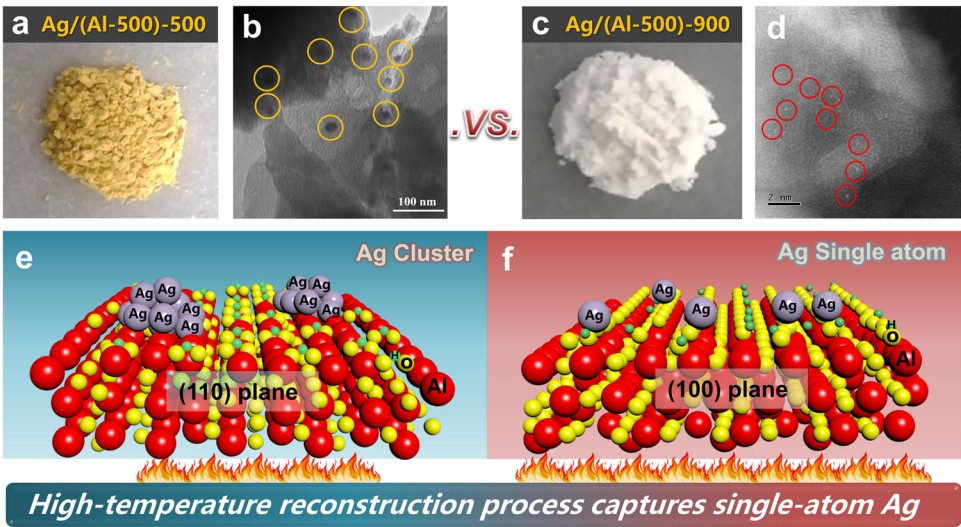

**Fig. 6 | Schematic diagram of high-temperature-induced crystal transition. a–d** Changes in color and particle size after calcination of Ag/(Al-500)-500 and Ag/(Al-500)-900 samples. **e–f** Schematic diagram illustrating the high-temperature capture process of Ag single atoms.

hydroxyl groups form on γ-Al₂O₃, providing ample anchor points for Ag to achieve single-atom dispersion. Furthermore, Ag single atoms captured at 900 °C on γ-Al₂O₃ exhibit superior catalytic activity and stability in the HC-SCR and O₃ decomposition processes. These findings offer innovative insights into designing thermally stable single atoms by creating additional anchor points for the capture of single atoms at high temperatures. This progress can stimulate further developments in the field of single-atom catalysis.

## Methods

### Preparation of catalyst support

During the experiment, we used pseudo-boehmite (AlOOH) as a precursor for γ-Al₂O₃. We prepared a series of γ-Al₂O₃ samples with different exposed crystal surfaces by calcining them in a muffle furnace at various temperatures, denoted as Al-X (where X indicates the calcination temperature, 500 °C, 600 °C, 700 °C, 800 °C and 900 °C).

### Preparation of Ag/(Al-X)

Supported Ag catalysts were prepared via the impregnation method, with Ag loadings of 1 wt%. In brief, we weighed an appropriate amount of catalyst support (Al-X) and added it to a silver nitrate (AgNO₃) solution in deionized water, stirring the mixture thoroughly. After thorough agitation for 2 h, we performed vacuum rotary evaporation at 60 °C to remove water. Subsequently, we dried the obtained samples in a constant-temperature oven at 120 °C for 6 h. The fully dried samples were then heated in a muffle furnace at a rate of 5 °C/min to 500 °C and calcined in air for 3 h. Finally, the resulting catalysts were denoted as Ag/(Al-X) (where X indicates the calcination temperature, 500 °C, 600 °C, 700 °C, 800 °C and 900 °C). The detailed preparation process is shown in Supplementary Fig. 11.

### Preparation of Ag/(Al-500)-X

An appropriate amount of catalyst support (Al-500) was added to a solution of silver nitrate (AgNO₃) in deionized water (Ag loading 1 wt%), stirring the mixture thoroughly. After thorough agitation for 2 h, we performed vacuum rotary evaporation at 60 °C to remove water. Subsequently, we dried the obtained samples in a constant-temperature oven at 120 °C for 6 h. The fully dried samples were then heated in a muffle furnace at a rate of 5 °C/min to 500 °C and 900 °C and calcined in air for 3 h. Finally, the resulting catalysts were denoted as Ag/(Al-500)-X (where X indicates the calcination temperature, 500 °C and 900 °C).

### Catalyst Characterization

X-ray diffraction (XRD): We characterized the phase structure of the catalyst using XRD. This was carried out using a D8 ADVANCE diffractometer from Bruker in Germany. We utilized Cu-Kα rays (λ = 0.15406 nm) with a tube voltage and current of 40 kV and 40 mA, respectively. The scanning range was 10–90°, the scanning rate was 6°·min⁻¹, and the scanning step was 0.02°. In situ XRD tests were performed while heating in air by powder XRD with a Malvern Panalytical XRD (Empyrean, Cu-Kα radiation source) in the 2 theta range of 10–90°. The specific voltage-current for in situ XRD was 45 kV, 40 mA. The temperature was increased from room temperature to 300 °C at 20°·min⁻¹, and in situ XRD tests were performed after a 10 min hold at 300 °C. The temperature was then increased to 400 °C at a rate of 5°·min⁻¹, and in situ XRD tests were performed after a 10 min hold. This procedure was followed up to 900 °C.

High Resolution Transmission Electron Microscopy (HR-TEM): HR-TEM allowed us to observe the microscopic morphology of the catalyst support, loaded active components, and particle size. This experiment was carried out using a JEOL-JEM 2010 with an accelerating voltage of 200 kV.

High-angle annular dark-field image-scanning transmission electron microscopy (HAADF-STEM): Experiments were performed using a JEOL-JEM-ARM 200F spherical differential electron microscope with an accelerating voltage of 200 kV.

X-ray Photoelectron Spectroscopy (XPS): the valence information of Ag species was characterized using XPS. The XPS measurements were recorded with a scanning X-ray microprobe (Thermo Fisher Scientific K-Alpha) using Al Kα radiation.

X-ray Absorption Fine Structure Spectroscopy (XAFS): XAFS is a powerful tool for analyzing the local structure of substances and can be divided into X-ray Absorption Near Edge Structure Spectroscopy (XANES) and Extended X-ray Absorption Fine Structure Spectroscopy (EXAFS), depending on the energy range measured. XANES is commonly used to analyze the valence and coordination chemistry of specific absorbing atoms, while EXAFS provides information such as the coordination number and coordination distance of the nearest neighbor of a specific absorbing atom. In this experiment, we measured the XANES and EXAFS spectra of the Ag K-edge at room temperature using transmission mode at the Shanghai Light Source (SSRF) BL14W1 line station. The storage ring energy was 3.5 GeV and the average storage ring current was 200 mA. Data acquisition and processing for XAFS were performed using Athena software, which is part

of the IFFEFIT software package. The XANES spectra were normalized based on the jump height of the spectra, and the first-order derivatives were used to analyze changes in energy at the absorption edges to determine the valence of the active components. The oscillation function $\chi(k)$ of the EXAFS was extracted through Spline Smoothing and then converted into an R-space spectrum using Fourier Transform with $k^2$ weighting.

$NH_3$ adsorption in situ diffuse reflectance FTIR spectroscopy (in situ DRIFTS): Hydroxyl groups (OH) on the catalyst surface can act as Brønsted acidic sites for $NH_3$ adsorption, and the number of hydroxyl groups in the catalyst can be deduced from the peak area of the hydroxyl depletion peak in the $NH_3$ adsorption in situ DRIFTS results. The experiment was conducted using the same instrument as used for FTIR transmission spectroscopy, equipped with an in situ reaction cell (Harrick) connected to a reaction gas control system for adsorption and purging, and a Thermo Spectra-Tech regulator to control the reaction temperature. The procedure involved placing 30 mg of catalyst in the in situ cell reactor, pretreating it with high-purity air for 30 min at 400 °C, and then cooling it down to 30 °C. The catalyst background was obtained by switching to a high-purity $N_2$ purge for 30 min. Subsequently, 500 ppm $NH_3$ was adsorbed for 0.5 h after deducting the background, followed by purging with high-purity $N_2$ for 15 min (resolution of 4 cm$^{-1}$, 64 scans).

### DFT calculations

ab initio molecular dynamics (AIMD): AIMD is a valuable method for studying the stability of atoms at high temperatures. To search for the equilibrium structure of the $Al_2O_3$ surface at different temperatures, we performed ab initio molecular dynamics calculations on the $Al_2O_3$ (110) surface within the framework of Spin-polarized density functional theory (DFT) calculations. The calculations were carried out using the LAMMPS software package for molecular dynamics simulations. The $Al_2O_3$ (110) configuration was designed, and a 25 ps relaxation was conducted using a canonical ensemble (NVT, where N is the number of atoms, V is the volume, and T is the temperature). The relaxation included a 500 K heat bath with a Nose-Hoover thermostat, followed by a quench to 300 K. During quenching, the density/volume of the structure was kept constant, and the four atomic layers at the bottom were fixed. Subsequently, the structure was allowed to fully relax for 40 ps at 0 K, 300 K, and 1173 K under a regular system. Finally, we combined the ab initio molecular dynamics results with thermodynamic concepts to deduce the structural evolution of $Al_2O_3$ during the warming process. The stability of the free surfaces of $Al_2O_3$ (110) and (100) in the presence of temperature was assessed by calculating the surface free energy, which is given by the formula:

$$\sigma = \frac{1}{2A}\left(\text{Eslab} + \text{ZPE} - TS - N_{Al}u_{Al} - N_o u_o\right) \quad (1)$$

where Eslab is the total energy after complete relaxation; $N_{Al}$ and $N_O$ are the numbers of Al and O in the supercell, respectively. $u_{Al}$ and $u_O$ are the chemical potentials of surface Al and O.

To further study the stability of single-atom Ag on the γ-$Al_2O_3$ (100) facet, ab initio molecular dynamics (AIMD) simulations were conducted. Three Ag atoms were adsorbed on the surface hydroxide radicals on (100) facets and relaxed using the same accuracy as described above. Based on the optimized structure, the adsorption models were heated to 773 K and balanced in an NVT ensemble for 10 ps with a time step of 1 fs. The energy convergence criterion was decreased to 3.67*10$^{-6}$ Hartree to accelerate the molecular dynamics calculation.

After determining the state of Ag atoms on different facets, two models that only contained a Ag cluster with 14 Ag atoms or a Ag single atom were built to study the electronic properties including the charge density difference, Bader charge and density of states. These models

were further optimized by the Vienna ab initio simulation package (VASP) to obtain accurate values. The Projector augmented wave method with a cutoff energy of 400 eV accompanied by Perdew-Burke-Ernzerhof functional was used in the DFT calculations. The DFT-D3 method was used to correct the influence of van der Waals interactions. The energy convergence criterion was 10$^{-5}$ eV and the force convergence criterion was 0.02 eV/Å, respectively. Brillouin zone integration was performed with the K-point mesh of the Γ point.

### Activity tests

We evaluated the $H_2$-assisted $C_3H_6$-SCR of NO activity of the catalyst in a fixed-bed flow reactor. The device uses a quartz tube reactor of Φ4 × 300 mm in size, placed in a resistance furnace with a constant temperature zone of approximately 150 mm. A Fourier transform infrared spectrometer (Nicolet iS50) with a 2 m light range gas cell was used to measure the concentrations of product components during the evaluation of the catalyst activity. The gas composition included a gas mixture flow rate of 100 mL·min$^{-1}$, with $C_3H_6$ at 1600 ppm, NO at 800 ppm, $H_2$ at 1%, $O_2$ at 10%, $N_2$ as the balance gas, and a gas volume velocity (GHSV) of 118,000 h$^{-1}$. The reaction temperature ranged from 150–450 °C. $NO_x$ conversion was calculated using the formula:

$$NO_x \text{conversion} = \left(\left[NO\right]_{in} - \left[NO\right]_{out} - \left[NO_2\right]_{out}\right)/\left[NO\right]_{in} \times 100\% \quad (2)$$

The ozone decomposition activity was tested by placing a 100 mg sample with a size of 40–60 mesh in a continuous flow quartz reactor (diameter of quartz reactor = 4 mm) at room temperature (30 °C). The total air flow was controlled at 1.4 L/min. The space velocity was 840 L·g$^{-1}$·h$^{-1}$. The ozone was generated by an ozone generator and controlled to 40 ± 2 ppm with an ozone detector (Model 202, 2B Technologies). The ozone concentrations at the inlet and outlet were measured as $C_{in}$ and $C_{out}$, respectively. The $O_3$ conversion rate was calculated according to the following Equation:

$$O_3 \text{conversation} = (C_{in} - C_{out})/C_{in} \times 100\% \quad (3)$$

## Data availability

The data that supporting the findings of this study are available within the paper and its supplementary information. All relevant data are provided in the Source Data file. Source data are provided with this paper.

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

## Acknowledgements

F.W. received financial support from the National Natural Science Foundation of China (No. 52370113, 52000093). F.W. received financial support from Yunnan Fundamental Research Projects (Grant No. 202101BE070001-001).

## Author contributions

J.L. synthesized the catalyst and performed the reactions, and drafted the manuscript. K.L. methodology, review the manuscript. Z.L. supported for XAS data analysis. C.W., Y.L., and Y.P. performed the BET and in situ XRD experiments. N.P. revised the manuscript. J.M. and F.W. conceived the project and assisted with editing the manuscript. H.H. revised the manuscript.

## Competing interests

The authors declare no competing interests.
