## [Peer Review File · Nature Communications]

Capture of Single Ag Atoms through High-Temperature-Induced Crystal Plane ReconstructionREVIEWER COMMENTS

Reviewer #1 (Remarks to the Author):

I noticed that the author's previous work (Nat. Commun. 2020, 11(1), 529) on anchoring mechanisms shows that the (100) crystalline face of γ -Al₂O₃ possesses more terminal hydroxyl groups than the (110) crystalline face and that Ag species are anchored to γ -Al₂O₃ mainly through terminal hydroxyl groups. In this manuscript, the author successfully prepared a thermally stable silver-based single-atom catalyst by building a large number of anchor sites through high-temperature-induced crystalline facet transformation. Specifically, the authors not only demonstrate the possibility that the crystallographic surface of γ -Al₂O₃ changes from (110) to (100) at high temperatures, but also solve the current thorny problem of monatomic susceptibility to sintering at high temperatures by digging deeper into the relationship between the crystallographic surface and the anchor point at high temperatures, which in turn regulates the dispersion of the metal. In my opinion, the idea of this work is very interesting, and could also lead to more new ideas for the design of single-atom. Hence, I recommend its publication with minor revision after considering the following concerns and making corresponding responses.

1. Why AlOOH was chosen as the precursor of γ -Al₂O₃ in this experiment? it is recommended to add the XRD spectra of fresh samples of AlOOH in the XRD part of this part of the high-temperature roasting crystal surface transformation, so as to better illustrate the process of the complete transformation of AlOOH to γ -Al₂O₃ after roasting.
2. In the Methods section, why was the choice made to roast AlOOH at different temperatures before loading Ag, instead of loading Ag first and roasting the catalyst at different temperatures?
3. The background section of the article is not sufficiently descriptive in relation to crystalline surfaces. Maybe this part can be improved, please add some more relevant research background in this section.
4. The hydroxyl anchoring of Ag on alumina proposed by the author is quite interesting, but the dominant reaction of single-atom Ag is limited. Has the author explored whether the anchoring mechanism of other supports or other metals is the same or comparable and whether there is a universal rule and corresponding dominant reaction?
5. I found too many decimal places in the Pore volume part of Supplementary Table 1. I would suggest to keep two decimal places. For example, 0.724 is corrected to 0.72 with two decimal places.
6. Ag loading should be shown in the caption of the figures, such as Fig. 3a. Trends in hydroxyl groups before and after Ag loading at different roasting temperatures.
7. Fig. 3 in the article is a little blurry when zoomed in, Is the DPI parameter setting problem or images inserted into the document being compressed? I suggest setting the DPI parameters a bit larger and removing the compression of the image in the document settings, then replacing it with a clearer one.
8. I saw that the method of roasting in the article did see changes in the content and type of hydroxyl group. However, whether the type and content of the hydroxyl group would change during the loading of Ag particles, and whether the Ag loading process would cause the reduction or increase of the hydroxyl group, how to verify or exclude this effect?

Reviewer #2 (Remarks to the Author):

In this manuscript, Wang and his co-workers successfully synthesized the Ag/AlOOH anti-sintering single atom catalyst. The γ -Al₂O₃ is heated at elevated temperature to generate more surface hydroxyl for anchoring Ag single atoms. The Ag single atoms formed at 900°C showed enhanced activity than that of 500°C for NO_x conversion. The report provides a new perspective and idea in the field of single atom catalysts and atomization of metal nanoparticles. I believe that this paper will attract considerable interest in the related community and that it can be published in Nature Communications after the authors addressing following issues:

1. The sentence of this manuscript is long and tedious. It should be simplified so that readers can

clearly understand the purpose and results of the research. Meanwhile, please pay attention to the writing standard, such as "1 wt.%, k2" and so on in Methods. What does the "1" mean in the "1Ag/AIOOH-900"? This is not an expression established by usage.

2、 Please provide a schematic diagram of the synthesis route to make the reader more clear synthesis method. The figures of this manuscript should be rearranged to make it more logic. It is tedious that the three main figures from figure 1. to figure 3. only tells the formation of OH on after high temperature treatment. They can certainly be arranged into one or two figures.

3、 The author emphasize the thermal stability of Ag/AIOOH-900 and it is well know that stability of catalyst is highly important for NO_x catalytic conversion of SCR. Is the Ag/AIOOH-900 stable under catalytic durability test?

4、 Did Ag content change before and after Ag nanoparticles(cluster) to Ag single atom transformation? (by ICP measurement)

5、 It is suggested that the authors supplement XPS test to analyze the oxidation state of Ag before and after monoatomization.

6、 Some relevant literature concerning the nanoparticle-to-single atom transformation should be cited (J. Am. Chem. Soc. 2023, 145, 17, 9540-9544, J. Am. Chem. Soc. 2023, 145, 17, 9540-9547).

Reviewer #3 (Remarks to the Author):

The work by Li et al reported the thermal treatment of Al₂O₃ to control the content of surface hydroxyl groups, which they proposed to anchor the Ag. Based on their former publication at Nature Communications volume 11, Article number: 529 (2020), this work should clearly address the following questions due to potential integrity issues.

-- In their former publication, they show that Al₂O₃ calcined at 500 degree is sufficient to form isolated Ag, and even can hold a higher Ag content. In stark contrast, in this new submission, they used a low Ag content of 1 wt% but stated that the Al₂O₃ should be calcined at a much higher temperature to have good dispersion. A strong clarification and justification for the discrepancy should be provided.

-- Terminology is wrong. From the XRD, the material is Al₂O₃, not the oxyhydroxide. "roasting" is also not typically used.

-- XPS survey and high resolution XPS should be provided for O, Ag, Al

-- the resolution of TEM is low. There are also significant amorphous areas in the images. In addition, TEM only shows small area that cannot reflect the whole structures. Indeed, in Fig 1e, there are also a large portion with shorter layer distance.

-- it is also unclear why surface hydroxide will favor the Ag adsorption. The extra hydrogen intuitively will repulse Ag, both of which are positively charged. If the proposed mechanism is correct, why not directly use AIOOH to adsorb Ag?

-- For XAFS data, the peak positions of samples are confusing (Fig5a). The peak position lies at lower energy compared to Ag, which indicates a lower valence. The impregnation was performed in the same for both Ag/AIOOH, but why the valance differs so much. For the EXAFS data, why there is only Ag-O bond but no Ag-Ag for Ag₂O standard. From shape and trend analysis, the peak and FT are similar for 1Ag/AIOOH-900 and Ag₂O.

Reviewer #4 (Remarks to the Author):

The manuscript offered by Li and co-workers outlined a straightforward approach involving carefully regulating the roasting temperature to introduce additional hydroxyl groups to AIOOH, thereby creating anchor sites for Ag species. The authors presented a series of basic characterizations for prepared AIOOH materials including HR-TEM, HAADF-STEM, FTIR, XAFS, DFT and AIMD simulation. In this way, the existing hypothesis and conclusion cannot be well reconciled. Additionally, the results presented herein may not significantly advance the field beyond the state-of-the-art. The referee considers this issue substantial and critical. More specifically, modification of AIOOH with additional hydroxyl groups is not in itself substantive enough to motivate publication, particularly as the principal technical and scientific conclusions

related to the Ag anchoring and its properties are not yet fully developed, consistent, or supported by the theory and the experiments, which is described in more detail below. Consequently, I cannot recommend this work in its current state for publication in Nature Communications.

1. The author alters the crystal plane of AlOOH upon calcination at varying temperatures, ranging from (110) to (100), with changes corresponding to increasing temperatures. The verification is solely conducted through HR-TEM, which shows regional limitations (Figure 1). It is crucial to note whether the (100) crystal plane is consistently identified across all locations for AlOOH- 900 °C. Additionally, in situ XRD results would be essential for a comprehensive illustration of the transformation in the crystalline structure.

2. Authors have emphasized the function of hydroxyl groups in anchoring Ag atoms. Through calcination, the hydroxyl groups on the surface of the AlOOH are changed. Specifically, whether there is an increase in the quantity of hydroxyl groups or a more uniform distribution, however, from the results presented here, it is not clear which influence factors are most dominant.

3. In addition to employing the basic DFT and AIMD simulation, the physics regarding the effect of high-temperature calcination on the formation of hydroxyl groups on the AlOOH need to be further clarified. The main reason for this alteration can be attributed to the controlled conditions of calcination, influencing the thermodynamics of hydroxyl group generation and their subsequent distribution on the material surface. The temperature, duration, and atmosphere during calcination play crucial roles in determining whether there is an augmentation in the total number of hydroxyl groups or a more even dispersion of these groups across the surface.

4. According to the image of 1Ag/AlOOH at different temperatures, the reviewer lacks information regarding whether the distribution of Ag is more even or dense. The XRD results indicate a lack of discernible difference or statistical significance. It is recommended to incorporate additional compelling experiments to strengthen the findings.

5. To establish the universality of the material structure, it is suggested to explore alternative systems for verification. It is recommended exam the performance of alumina-supported silver (Ag/Al₂O₃) catalysts in different catalytic reactions or under varied conditions to provide a more comprehensive assessment of its versatility and applicability.

6. Before such an obvious issue is well resolved, the current manuscript does not allow this referee to endorse it for possible publication on Nat Commun. The manuscript requires a major revision. Please see some minor comments below to improve the manuscript:

- 1) Introduction was poorly drafted: it has both focus issue and logic issue. Re-writing is necessary.
- 2) The abscissa of Fig.3d shown "wavenumbers" is totally wrong.
- 3) The corresponding crystal plane is not marked in Figure 1c, d.
- 4) There is no definition or standard abbreviation for the position where "HC-SCR" appears for the first time in the manuscript.

RESPONSE TO REVIEWERS' COMMENTS

Reviewer #1 (Remarks to the Author):

I noticed that the author's previous work (Nat. Commun. 2020, 11(1), 529) on anchoring mechanisms shows that the (100) crystalline face of γ -Al₂O₃ possesses more terminal hydroxyl groups than the (110) crystalline face and that Ag species are anchored to γ -Al₂O₃ mainly through terminal hydroxyl groups. In this manuscript, the author successfully prepared a thermally stable silver-based single-atom catalyst by building a large number of anchor sites through high-temperature-induced crystalline facet transformation. Specifically, the authors not only demonstrate the possibility that the crystallographic surface of γ -Al₂O₃ changes from (110) to (100) at high temperatures, but also solve the current thorny problem of monatomic susceptibility to sintering at high temperatures by digging deeper into the relationship between the crystallographic surface and the anchor point at high temperatures, which in turn regulates the dispersion of the metal. In my opinion, the idea of this work is very interesting, and could also lead to more new ideas for the design of single-atom. Hence, I recommend its publication with minor revision after considering the following concerns and making corresponding responses.

Response: We thank the referee for the positive evaluation of our work. Our point-by-point responses are listed below.

#1-1. Why AlOOH was chosen as the precursor of γ -Al₂O₃ in this experiment? it is recommended to add the XRD spectra of fresh samples of AlOOH in the XRD part of this part of the high-temperature roasting crystal surface transformation, so as to better illustrate the process of the complete transformation of AlOOH to γ -Al₂O₃ after roasting.

Response: We sincerely appreciate your comments. AlOOH, as a commonly used precursor for γ -Al₂O₃, has low cost, a large specific surface area, and abundant hydroxyl content, so we chose AlOOH as the precursor of γ -Al₂O₃. In addition, we also further confirmed the transformation of AlOOH to γ -Al₂O₃ by in situ XRD; we

added **Figure R1 as Supplementary Fig. 1, Figure R2 as Supplementary Fig. 2** in the revised Supplementary Information.

The revision in the manuscript (page 6, line 93-100)

The XRD pattern of fresh AlOOH and in situ XRD were used to demonstrate the crystal transition process of AlOOH calcined at different temperatures, as shown in Supplementary Fig. 1-2. The AlOOH crystalline phase persists at calcination temperatures up to 300 °C and starts taking on the crystalline form of γ -Al₂O₃ at 400 °C. Diffraction peaks at 37.5°, 45.7°, 60.5°, 66.6°, and 84.5° were observed, corresponding to the γ -Al₂O₃ (311), (400), (511), (440), and (444) crystal planes (JCPDS 02-1420), which indicates that AlOOH is converted to γ -Al₂O₃ starting at about 400 °C.

Figure R1 (Supplementary Fig. 1). XRD profiles of fresh AlOOH.

Figure R2 (Supplementary Fig. 2). In situ XRD profiles of AlOOH calcined at different temperatures.

#I-2. In the Methods section, why was the choice made to roast AlOOH at different temperatures before loading Ag, instead of loading Ag first and roasting the catalyst at different temperatures?

Response: We sincerely appreciate your comment. The reasons we chose to calcine before loading Ag rather than load Ag before calcining are as follows: Firstly, the crystallization of AlOOH in the initial state is incomplete, and secondly, we aimed to use γ -Al₂O₃ with different hydroxyl contents to load Ag. Therefore, we had to load Ag after converting AlOOH into γ -Al₂O₃ with different hydroxyl contents by calcining AlOOH at different temperatures first.

#I-3. The background section of the article is not sufficiently descriptive in relation to crystalline surfaces. Maybe this part can be improved, please add some more relevant research background in this section.

Response: Thanks for your helpful suggestions. The relevant literature has been added to the revised manuscript.

The revision in the manuscript (page 5, line 74-77)

Hu et al¹⁸. loaded Pd on different crystal facets of CeO₂ and found that on the CeO₂ (100) facets, Pd exists predominantly in the form of Pd SAs (single atoms). In contrast, on the CeO₂ (111) facets, Pd readily aggregates into Pd clusters.

18. Hu, B. et al. Distinct Crystal-Facet-Dependent Behaviors for Single-Atom Palladium-On-Ceria Catalysts: Enhanced Stabilization and Catalytic Properties. *Adv. Mater.* **34**, 2107721 (2022).

#1-4. The hydroxyl anchoring of Ag on alumina proposed by the author is quite interesting, but the dominant reaction of single-atom Ag is limited. Has the author explored whether the anchoring mechanism of other supports or other metals is the same or comparable and whether there is a universal rule and corresponding dominant reaction?

Response: We genuinely thank you for your valuable suggestions. Actually, we have investigated the universality of the “terminal hydroxyl group anchoring mechanism” to other supports (CeO₂) and other metals (Cu, Ni, Mn, Fe, Co, Pt, and Pd) in our newly published study. This anchoring mechanism has guided the design and synthesis of a series of catalysts that exhibit excellent performance in their corresponding catalytic reactions. Based on the “terminal hydroxyl group anchoring mechanism,” we propose a strategy of “pre-occupied anchoring-site” (*Appl. Catal. B: Environ.* 2024, 344, 123655) by using an inexpensive transition metal, Cu, which has a high anchoring strength to Al₂O₃, to preoccupy the anchors on the Al₂O₃ surface (**Figure R3**). As a result, due to the lack of anchoring sites, Ag can be forced to agglomerate into AgNPs, which greatly improves the conversion of NH₃. This “pre-occupation of anchorage sites” strategy is widely applicable to a variety of inexpensive transition metals such as Mn, Co, Ni, and Fe, which can pre-occupy the Ag anchoring sites, forcing the more expensive Ag to aggregate into Ag nanoparticles, lowering the cost and increasing the ammonia oxidation activity at the same time (**Figure R4**).

In addition, we prepared different morphologies of CeO₂, a reducible oxide support, (nanocubes, nanoparticles, and nanorods) and found that the abundant terminal OH groups on CeO₂-NC support induce monoatomic dispersion of Ag atoms (*Angew. Chem. Int. Ed.* 2024, e202318492). On the other hand, due to the lack of terminal hydroxyl groups on CeO₂-NP and CeO₂-NR, Ag tends to aggregate into

nanoparticles rather than exhibit single-atom dispersion. Finally, the anchoring mechanism based on terminal hydroxyl groups can also be applied to other metals. Pt and Pd can be anchored to the CeO₂ (100) surface in the form of dispersed monometallic atoms through terminal hydroxyl groups. Atomically dispersed Pt/Pd exhibits morphology- and temperature-dependent CO selectivity in the catalytic CO₂ hydrogenation reaction (**Figure R5**).

The revision in the manuscript (page 4, line 53-64):

Based on the proposed “terminal hydroxyl group anchoring mechanism,” we constructed more effective NH₃-SCO active sites using the “pre-occupied anchoring-site” strategy¹⁶. We used Cu with stronger anchoring strength to pre-occupy the anchoring sites to force Ag agglomeration, resulting in the construction of an efficient NH₃-SCO catalyst at low Ag loading. Meanwhile, we found that the “terminal hydroxyl group anchoring mechanism” is also applicable to the anchoring of other non-precious metals (Fe, Co, Ni, and Mn) on Al₂O₃ support. In addition, we found that on the CeO₂ support, the terminal hydroxyl group is also an anchoring site for Ag atoms, and terminal OH group on the CeO₂ (100) surface can firmly anchor Ag via the formation of a dumbbell structure¹⁷. Moreover, other metals (Pt, Pd, etc.) on CeO₂ can also be directly anchored to terminal hydroxyl group.

Figure R3. Anchoring of Cu on Al₂O₃.

Figure R4. DFT calculation results for the anchoring capability of Co, Mn, Fe, or Ni on the γ -Al₂O₃ (100) surface.

Figure R5. Anchoring of Ag on CeO₂ NP, CeO₂ NC, and CeO₂ NR, respectively.

#I-5. I found too many decimal places in the Pore volume part of Supplementary Table 1. I would suggest to keep two decimal places. For example, 0.724 is corrected to 0.72 with two decimal places.

Response: We sincerely appreciate your advice to adjust the decimal places. We revised Supplementary Table 1 as follows:

Revised Supplementary Table 1.

Sample	S_{BET} ($\text{m}^2 \text{g}^{-1}$)	Pore diameter (d) (nm)	Pore volume (V) ($\text{cm}^3 \text{g}^{-1}$)
Ag/(Al-500)	325.5	3.9	0.72
Ag/(Al-600)	289.7	6.6	0.74
Ag/(Al-700)	255.4	6.6	0.73
Ag/(Al-800)	231.3	7.8	0.69
Ag/(Al-900)	182.4	7.9	0.55

#1-6. Ag loading should be shown in the caption of the figures, such as Fig. 3a. Trends in hydroxyl groups before and after Ag loading at different roasting temperatures.

Response: We sincerely appreciate your advice, we revised the caption of the original Fig. 3a in the revised layout of Fig. 2a (**Figure R6 (Fig. 2)**).

Figure R6 (Fig. 2). Hydroxyl change characterization and Ag anchoring mechanism schematic diagram. **a** Trends in hydroxyl groups before and after Ag loading at different calcination temperatures (Ag loadings (0, 1%)). **b** Resolution of OH consumption peaks after in situ DRIFTS of NH₃ adsorption over AlOOH at different temperatures. **c** Resolution of OH consumption peaks after in situ DRIFTS of NH₃ adsorption over Ag/(Al-500 °C, 600 °C, 700 °C, 800 °C, 900 °C) at different temperatures. **d** Terminal OH contents before and after Ag loading determined through in situ DRIFTS measurements at different calcination temperatures. **e** Schematic illustration of the Ag anchoring mechanism.

#1-7. Fig. 3 in the article is a little blurry when zoomed in, Is the DPI parameter setting problem or images inserted into the document being compressed? I suggest setting the DPI parameters a bit larger and removing the compression of the image in the document settings, then replacing it with a clearer one.

Response: We sincerely appreciate your advice. We set the DPI parameter to 1000, and revised the original Fig.3 as follows:

Figure R7 (Fig. 2). Hydroxyl change characterization and Ag anchoring mechanism.

#1-8. I saw that the method of roasting in the article did see changes in the content and type of hydroxyl group. However, whether the type and content of the hydroxyl group would change during the loading of Ag particles, and whether the Ag loading process would cause the reduction or increase of the hydroxyl group, how to verify or exclude this effect?

Response: We sincerely appreciate your comment. In order to verify the effect of the Ag loading process on the hydroxyl content, we examined the difference in hydroxyl content before and after water impregnation by in situ DRIFTS spectra of NH_3 adsorption of samples prepared from AlOOH calcined at different temperatures. As shown in **Figure R8**, there is no difference in terminal hydroxyl content (3788 cm^{-1}) before and after water impregnation of AlOOH calcined at either $500\text{ }^\circ\text{C}$ or $900\text{ }^\circ\text{C}$. The terminal hydroxyl group serves as an anchoring site for Ag on $\gamma\text{-Al}_2\text{O}_3$, and the water impregnation process has little effect on the content of terminal hydroxyl groups, so we can rule out the effect of the Ag impregnation loading process on the hydroxyl groups.

Figure R8. Hydroxyl change before and after water impregnation.

Reviewer #2 (Remarks to the Author):

In this manuscript, Wang and his co-workers successfully synthesized the Ag/AlOOH anti-sintering single atom catalyst. The γ -Al₂O₃ is heated at elevated temperature to generate more surface hydroxyl for anchoring Ag single atoms. The Ag single atoms formed at 900 °C showed enhanced activity than that of 500 °C for NO_x conversion. The report provides a new perspective and idea in the field of single atom catalysts and atomization of metal nanoparticles. I believe that this paper will attract considerable interest in the related community and that it can be published in Nature Communications after the authors addressing following issues:

Response: We thank the referee for the positive evaluation of our work. Our point-by-point responses are listed below.

#2-1. The sentence of this manuscript is long and tedious. It should be simplified so that readers can clearly understand the purpose and results of the research. Meanwhile, please pay attention to the writing standard, such as “1 wt.%, k²” and so on in Methods. What does the “1” mean in the “1Ag/AlOOH-900”? This is not an expression established by usage.

Response: We sincerely thank the reviewer for pointing out these issues. We have modified the sentence and standardized the expression according to your suggestions in the revised manuscript. You mentioned, “What does the “1” mean in the “1Ag/AlOOH-900”? “1” represents a 1 wt% loading of Ag in this manuscript, but for better readability, we have modified the method section with a detailed description, and the sample name has been changed to Ag/(Al-X), where X indicates the calcination temperature (500 °C, 600 °C, 700 °C, 800 °C, and 900 °C) of AlOOH. The detailed preparation process is shown in **Figure R9** (Supplementary Fig. 11).

The revision in the manuscript (page 25-26, line 411, 422)

1 wt%

The revision in the manuscript (page 28, line 468)

k² weighting

The revision in the manuscript (page 29, line 495-496)

ab initio molecular dynamics

The revision in the manuscript (page 25, line 403-419)

Preparation of catalyst support

During the experiment, we used pseudo-boehmite (AlOOH) as a precursor for γ -Al₂O₃. We prepared a series of γ -Al₂O₃ samples with different exposed crystal surfaces by calcining them in a muffle furnace at various temperatures, denoted as Al-X (where X indicates the calcination temperature, 500 °C, 600 °C, 700 °C, 800 °C and 900 °C).

Preparation of Ag/(Al-X)

Supported Ag catalysts were prepared via the impregnation method, with Ag loadings of 1 wt%. In brief, we weighed an appropriate amount of catalyst support (Al-X) and added it to a silver nitrate (AgNO₃) solution in deionized water, stirring the mixture thoroughly. After thorough agitation for 2 h, we performed vacuum rotary evaporation at 60 °C to remove water. Subsequently, we dried the obtained samples in a constant-temperature oven at 120 °C for 6 h. The fully dried samples were then heated in a muffle oven at a rate of 5 °C/min to 500 °C and calcined in air for 3 h. Finally, the resulting catalysts were denoted as Ag/(Al-X) (where X indicates the calcination temperature, 500 °C, 600 °C, 700 °C, 800 °C and 900 °C). The detailed preparation process is shown in Supplementary Fig. 11.

#2-2. Please provide a schematic diagram of the synthesis route to make the reader more clear synthesis method. The figures of this manuscript should be rearranged to make it more logic. It is tedious that the three main figures from figure 1. to figure 3. only tells the formation of OH on after high temperature treatment. They can certainly be arranged into one or two figures.

Response: Thank you for your valuable comments. We provided a schematic diagram of the synthesis route in the revised SI (Figure R9 as Supplementary Fig. 11) and changed the layout of the original manuscript's first 3 figures (Figure R10 as Fig. 1, Figure R11 as Fig. 2).

Figure R9 (Supplementary Fig. 11). Catalyst synthesis route.

Figure R10 (Fig. 1). The crystal plane transformation process of AlOOH calcined at different temperatures, and AIMD and DFT calculations.

Figure R11 (Fig. 2). Hydroxyl change characterization and Ag anchoring mechanism

#2-3. The author emphasize the thermal stability of Ag/AlOOH-900 and it is well know that stability of catalyst is highly important for NO_x catalytic conversion of SCR. Is the Ag/AlOOH-900 stable under catalytic durability test?

Response: We sincerely appreciate your advice. We tested the stability of the Ag/(Al-900) sample for C₃H₆-SCR, as shown in **Figure R12 (Fig. 5)**. Within 50 hours, approximately 100% and 75% NO_x conversion rates can be maintained at reaction temperatures of 350 °C and 300 °C, respectively. In addition, we have confirmed the

thermal stability of Ag single-atom samples (Ag/(Al-900)) by *ab initio* molecular dynamics simulation. As shown in Figure R11b-c (Fig. 5b-c), during the observation time of 10,000 fs at 723 K, Ag atoms were stably anchored on the surface of γ -Al₂O₃ (100) without agglomeration, suggesting the good thermal stability of Ag/(Al-900).

The revision in the manuscript (page 21, line 327-340)

A catalyst's stability is another essential indicator of its performance. For the Ag/(Al-900) sample, Fig. 5c displays NO_x conversion of about 100% and 75% for 50 hours at reaction temperatures of 350 °C and 300 °C, respectively. Subsequently, we further verified the stability of Ag/(Al-900) samples through *ab initio* molecular dynamics simulation. As shown in Supplementary Figure 10, the system is stable both in terms of temperature and potential energy. We also calculated the Ag-O and Ag-Ag atomic distances (Fig. 5d), and the results showed that the average distance between Ag and the surrounding O atoms is always shorter than Ag-O bonds within Ag₂O crystals. In addition, the distance between two Ag atoms is always longer than the Ag-Ag bond in the bulk, which indicates that the terminal hydroxyl groups on the γ -Al₂O₃ surface can stably anchor Ag single atoms without agglomeration. Therefore, during the simulation time of 10000 fs, Ag can be stably anchored on the γ -Al₂O₃ (100) surface at 773 K while maintaining the single-atom form (Fig. 5e), suggesting the good thermal stability of Ag/(Al-900).

Figure R12 (Fig. 5). Stability test and *ab initio* molecular dynamics simulation of Ag/(Al-900) catalysts.

#2-4. Did Ag content change before and after Ag nanoparticles(cluster) to Ag single atom transformation? (by ICP measurement)

Response: We sincerely appreciate your advice to characterize the Ag content change before and after the Ag nanoparticle (cluster) to Ag single-atom transformation. We characterized the Ag content by ICP measurement as shown in **Figure R13 (Supplementary Fig. 6)**. Clearly, the Ag content of the Ag cluster (Ag/(Al-500)) catalyst is 0.91 wt% by ICP measurement, which is similar to the Ag content of the Ag single-atom catalyst (Ag/(Al-900)) (0.91 wt%). We added the Ag content of the Ag/(Al-500) and Ag/(Al-900) samples in the revised Manuscript.

The revision in the manuscript (page 14, line 221-222)

ICP results indicated that the Ag content for both the Ag/(Al-500) and Ag/(Al-900) samples was 0.91 wt% (Supplementary Fig. 6).

Figure R13 (Supplementary Fig. 6). ICP measurement of Ag content of Ag/(Al-500) and Ag/(Al-900) samples.

#2-5. It is suggested that the authors supplement XPS test to analyze the oxidation state of Ag before and after monoatomization.

Response: Thanks for your suggestion. As shown in **Figure R14 (Supplementary Fig. 7)**, for the XPS spectra at the Ag 3d core level, the intensity of the Ag/(Al-900) sample is stronger compared to the Ag/(Al-500) sample, indicating that a large number of terminal hydroxyl groups on the surface of the sample induces single-atom Ag dispersion, resulting in an increase in the number of Ag atoms on the surface of the region. Secondly, the change in calcination temperature also caused a change in the valence state of Ag. The Ag 3d 5/2 peak of the Ag/(Al-500) sample is located at 368.2 eV, which is close to the position of the metallic silver energy peak. The binding energy of Ag in the Ag/(Al-900) sample shifted significantly to 368.7 eV with increasing calcination temperature, suggesting that the Ag species in the Ag/(Al-900) sample existed mainly in the oxidized state (Ag⁺). This result is also consistent with the XAFS results in the manuscript.

The revision in the manuscript (page 16-17, line 253-263)

We can observe that for the XPS spectra at the Ag 3d core level, the intensity of the Ag/(Al-900) sample is stronger compared to the Ag/(Al-500) sample, indicating that

the large number of terminal hydroxyl groups on the surface of the sample induces greater Ag dispersion, resulting in an increase in the number of Ag atoms on the surface in the region. Secondly, the change in calcination temperature also caused a change in the valence state of Ag. The Ag 3d 5/2 peak of the Ag/(Al-500) sample is located at 368.2 eV, which is close to the position of the metallic silver energy peak. The binding energy of Ag in the Ag/(Al-900) sample shifted to 368.7 eV, suggesting that the Ag species in the Ag/(Al-900) sample existed mainly in the oxidized state (Ag^+)^{29, 30}.

Figure R14 (Supplementary Fig. 7). XPS Ag 3d spectra of Ag/(Al-500) and Ag/(Al-900) samples.

#2-6. Some relevant literature concerning the nanoparticle-to-single atom transformation should be cited (J. Am. Chem. Soc. 2023, 145, 17, 9540-9547).

Response: Thank you for bringing up the recently published articles on single-atom catalysis. We have carefully reviewed the articles you mentioned. This article provides new perspectives on the precise control of single atoms to optimize catalytic

performance. We have incorporated these references into our manuscript to supplement and improve our discussion on single-atom catalysis. We cite the references in the revised manuscript as follows:

The revision in the manuscript (page 23, line 369-372)

Single-atom catalysts gained favor among researchers due to lower precious metal loadings, efficient atomic utilization, and well-defined active centre⁴⁹⁻⁵³. For precise control of single atoms to optimize catalytic performance, various methods have been proposed. In addition to enhancing strong metal-support interaction, trapping single atoms on defect-rich metal oxides⁵⁴⁻⁵⁵, stabilisation of single atoms by MOF structures⁵⁶ and through specific strategies such as laser ablation⁵⁷.

53. Liang, X., Fu, N. Yao, S. Li, Z. & Li, Y. The Progress and Outlook of Metal Single-Atom-Site Catalysis. *J. Am. Chem. Soc.* **144**, 18155-18174 (2022).

56. Liu, Y., et al. Fabricating polyoxometalates-stabilized single-atom site catalysts in confined space with enhanced activity for alkynes diboration. *Nat. Commun.* **12**, 4205 (2021).

57. Fu, N. et al. Controllable Conversion of Platinum Nanoparticles to Single Atoms in Pt/CeO₂ by Laser Ablation for Efficient CO Oxidation. *J. Am. Chem. Soc.* **145**, 9540-9547 (2023).

Reviewer #3 (Remarks to the Author):

The work by Li et al reported the thermal treatment of Al₂O₃ to control the content of surface hydroxyl groups, which they proposed to anchor the Ag. Based on their former publication at Nature Communications volume 11, Article number: 529 (2020), this work should clearly address the following questions due to potential integrity issues.

Response: We thank the referee for the positive evaluation of our work. Our point-by-point responses are listed below.

#3-1. In their former publication, they show that Al₂O₃ calcined at 500 degree is sufficient to form isolated Ag, and even can hold a higher Ag content. In stark contrast, in this new submission, they used a low Ag content of 1 wt% but stated that the Al₂O₃ should be calcined at a much higher temperature to have good dispersion. A strong clarification and justification for the discrepancy should be provided.

Response: We sincerely appreciate your comment. Firstly, we apologize for the lack of clarity in the experimental section, which led to the reviewer's misunderstanding of the experimental methods in this manuscript. In fact, this manuscript does not contradict the former publication; the calcination temperature after Ag loading is 500 °C in both this manuscript and the previous one (Nat. Commun. 2020, 11(1), 529). For ease of understanding, we have provided a synthesis route for the samples (**Figure R15 (Supplementary Fig. 11)**) as well as a detailed description in the methods section.

Specifically, 900 °C in this manuscript represents our calcination temperature for the Al₂O₃ precursor AlOOH. We controlled the calcination temperature to obtain γ -Al₂O₃ with different hydroxyl contents and then loaded it with Ag, and the final samples were all calcined at 500 °C.

Finally, our previous work has pointed out that the terminal hydroxyl group is the anchoring site for Ag and that terminal hydroxyl groups are more abundant on the (100) face, and many studies have been done on the universality of the terminal hydroxyl group anchoring mechanism to other metals and other supports (Nat. Commun. 2020, 11(1), 529. Appl. Catal., B. 2024, 344, 123655. Angew. Chem. Int. Ed. 2024, e202318492). On this basis, we found that the exposed crystalline surface of the support also plays an important role in the dispersion of the active components. Due to the fact that the surface energy of the crystalline surface changes during heat treatment, in this work, we exposed a larger number of terminal hydroxyl groups by means of high-temperature calcination-induced crystal facet changes, allowing the monoatomic dispersion of Ag.

Figure R15 (Supplementary Fig. 11). Catalyst synthesis route.

#3-2. Terminology is wrong. From the XRD, the material is Al_2O_3 , not the **oxyhydroxide**. “roasting” is also not typically used.

Response: We are really sorry for our careless mistakes. Thanks for your reminder. We have carefully checked the full text and changed the incorrect terminology for Al_2O_3 and also replaced “roasting, roast” with “calcining, calcine” in the revised manuscript.

#3-3. XPS survey and high resolution XPS should be provided for O, Ag, Al.

Response: We sincerely appreciate your recommendation to supplement the XPS test. The XPS data for O, Al, and Ag are shown in **Figure R16 (Supplementary Fig. 7)**. In the O 1s spectrum, peaks located at around 531.1 eV and 532.2 eV are detected, respectively, with the former attributed to lattice oxygen (O_L) while the latter value can be assigned to surface OH (O_OH) species. Clearly, we can see that the proportion of O_OH decreases from 41% in the Ag/(Al-500) sample to 30% in the Ag/(Al-900) sample, indicating that the total OH content is decreasing as the calcination temperature of AlOOH is increased, which is also consistent with our in situ DRIFTS results. In the Al 2p spectrum, the peaks for the Ag/(Al-500) and Ag/(Al-900) samples are located at 74.3 eV and 74.6 eV, respectively, which correspond to the 2p

peaks of the oxidized state of Al_2O_3 , confirming that the calcined AlOOH was indeed converted to Al_2O_3 . Furthermore, the Ag 3d spectra were used to reflect the valence information of Ag species. We can observe that the XPS intensity of the Ag/(Al-900) sample is increased compared to the Ag/(Al-500) sample, indicating that a large number of terminal hydroxyl groups on the surface of the sample induces Ag dispersion, resulting in an increase in the number of Ag atoms on the surface. Secondly, the change in calcination temperature also caused a change in the valence state of Ag. The Ag 3d 5/2 peak of the Ag/(Al-500) sample is located at 368.2 eV, which is close to the position of the metallic silver energy peak. The binding energy of Ag in the Ag/(Al-900) sample shifted significantly to 368.7 eV with increasing calcination temperature, suggesting that the Ag species in the Ag/(Al-900) sample existed mainly in the oxidized state (Ag^+). This result is also consistent with the XAFS results in the manuscript.

The revision in the manuscript (page 16-17, line 244-263)

Supplementary Fig. 7 shows XPS data for O, Al, and Ag. In the O 1s spectrum, peaks located at around 531.1 eV and 532.2 eV were detected, respectively, with the former attributed to lattice oxygen (O_L) and the latter value assigned to surface OH (O_OH) species^{26,27}. Clearly, we can see that the proportion of O_OH decreases from 41% in the Ag/(Al-500) sample to 30% in the Ag/(Al-900) sample, indicating that the total OH content decreases as the calcination temperature of AlOOH is increased, which is also consistent with our in situ DRIFTS results. In the Al 2p spectrum, the peaks for the Ag/(Al-500) and Ag/(Al-900) samples are located at 74.3 eV and 74.6 eV, respectively, which correspond to the 2p peaks of the oxidized state of Al_2O_3 ²⁸, confirming that the AlOOH was indeed converted to Al_2O_3 after calcination. We can observe that for the XPS spectra at the Ag 3d core level, the intensity of the Ag/(Al-900) sample is stronger compared to the Ag/(Al-500) sample, indicating that the large number of terminal hydroxyl groups on the surface of the sample induces greater Ag dispersion, resulting in an increase in the number of Ag atoms on the surface in the region. Secondly, the change in calcination temperature also caused a

change in the valence state of Ag. The Ag 3d 5/2 peak of the Ag/(Al-500) sample is located at 368.2 eV, which is close to the position of the metallic silver energy peak. The binding energy of Ag in the Ag/(Al-900) sample shifted to 368.7 eV, suggesting that the Ag species in the Ag/(Al-900) sample existed mainly in the oxidized state (Ag^+)^{29, 30}.

Figure R16 (Supplementary Fig. 7). XPS spectra of Ag/(Al-500) and Ag/(Al-900) samples. a O 1s. b Al 2p. c Ag 3d.

#3-4. the resolution of TEM is low. There are also significant amorphous areas in the images. In addition, TEM only shows small area that cannot reflect the whole structures. Indeed, in Fig 1e, there are also a large portion with shorter layer distance.

Response: Thank you for pointing this out. We re-characterized the lattice fringes of ALOOH calcined at different temperatures, and the HR-TEM results are shown in **Figure R17 (Fig. 1)**. In our re-supplied images, we have ensured that there are multiple clear lattice fringe regions in almost every image, as well as improved resolution. Furthermore, we additionally provided HR-TEM images of other regions of the Al-500 and Al-900 samples (**Figure R18**). As you mentioned, “TEM only shows a small area that cannot reflect the whole structure;” this is indeed a general limitation of TEM characterization. Therefore, we have also verified the findings in the manuscript by AIMD. The results showed that the $\gamma\text{-Al}_2\text{O}_3$ (110) surface underwent significant structural changes at this high temperature (Fig. 1g-h). Atoms on the surface rearranged to form an atomic arrangement similar to that on the $\gamma\text{-Al}_2\text{O}_3$ (100) crystal surface, indicating a tendency for the (110) surface to transform

into the (100) surface under high-temperature conditions. Additionally, surface free energy calculations using density functional theory (DFT) revealed that the surface free energy of $\gamma\text{-Al}_2\text{O}_3$ (110) increased with rising calcination temperature, while that of $\gamma\text{-Al}_2\text{O}_3$ (100) decreased with increasing temperature (Fig. 1i). The surface energy of $\gamma\text{-Al}_2\text{O}_3$ (100) at 1173 K was significantly lower than that of $\gamma\text{-Al}_2\text{O}_3$ (110), further supporting the transformation of the (110) surface into the (100) surface at elevated temperatures.

Figure R17 (Fig. 1). The crystal plane transformation process of AlOOH calcined at different temperatures, and AIMD and DFT calculations. **a-e** Al-500 °C, 600 °C, 700 °C, 800 °C, 900 °C. **f** $\gamma\text{-Al}_2\text{O}_3$ (110) and **g** $\gamma\text{-Al}_2\text{O}_3$ (100) crystal surface structures. **h** $\gamma\text{-Al}_2\text{O}_3$ (110) crystal surface structure at 1173 K. **i** Surface free energy

of $\gamma\text{-Al}_2\text{O}_3$ (110) and $\gamma\text{-Al}_2\text{O}_3$ (100) structures at different temperatures (0 K, 300 K and 1173 K).

Figure R18. Additional supplementary HR-TEM images for Al-500 and Al-900 samples. a, b Al-500. c, d Al-900.

#3-5. it is also unclear why surface hydroxide will favor the Ag adsorption. The extra hydrogen intuitively will repulse Ag, both of which are positively charged. If the proposed mechanism is correct, why not directly use AlOOH to adsorb Ag?

Response: We sincerely thank you for the important comment. Actually, during catalyst impregnation, the positively charged Ag is attached to the negatively charged O by electrostatic adsorption, and the positively charged H is attached to the O to form an OH group. Based on our previous work, we pointed out the terminal hydroxyl anchoring mechanism, and in order to better understand the anchoring mechanism, we provided a schematic diagram to illustrate the terminal hydroxyl anchoring mechanism in detail, as shown in **Figure R19 (Fig. 2)**. According to the difference between the coordination numbers of O and Al, we refer to one O atom coordinated to

one Al atom as terminal-OH (Type1), one O atom coordinated to two Al atoms as doubly bridging-OH (Type2), and one O atom coordinated to three Al atoms as triply bridging-OH (Type3).

Regarding the question, “If the proposed mechanism is correct, why not directly use AlOOH to adsorb Ag?” Firstly, AlOOH is not a stable support, and most studies have shown that at 350 or 400 °C, AlOOH decomposes and converts to γ -Al₂O₃. Secondly, although AlOOH itself has a high water content, most of its internal hydroxyl groups are coupled hydroxyl groups, which have little anchoring effect on active species. Therefore, it is not suitable to directly use AlOOH to adsorb Ag.

The revision in the manuscript (page 12, line 202-204)

The substantial reduction in terminal hydroxyl content following Ag loading reinforces the notion that Ag primarily anchors to γ -Al₂O₃ through terminal hydroxyl groups (Fig. 2e)

Figure R19 (Fig. 2). Ag anchoring mechanism schematic diagram.

#3-6. For XAFS data, the peak positions of samples are confusing (Fig5a). The peak position lies at lower energy compared to Ag, which indicates a lower valence. The impregnation was performed in the same for both Ag/AlOOH, but why the valance differs so much. For the EXAFS data, why there is only Ag-O bond but no Ag-Ag for

Ag₂O standard. From shape and trend analysis, the peak and FT are similar for 1Ag/AlOOH-900 and Ag₂O.

Response: Thanks for your comments. We apologize for any confusion caused by the XAF data fitted in our original manuscript. In X-ray Absorption Near Edge Structure (XANES), the position of the absorption edge corresponds to the absorption of X-rays by the sample at a specific energy, and this position usually varies with the valence state of the sample. Normally, a higher valence state corresponds to a higher energy position in the XANES spectrum, such that the XANES position should be higher for Ag⁺ than Ag⁰. However, there are some differences in practice, and the XANES positions of silver in the oxidized state are lower than in the metallic state. This is because although Ag⁺ has a higher valence state, its electron cloud is more tightly distributed and the shielding effect of the electron cloud is stronger, making the core electrons easier to stimulate, which causes the absorption edge position to move towards lower energies. So, the XANES position of silver in the oxidized state is lower than that of silver in the metallic state. Therefore, we have re-added the AgNO₃ standard for the Ag-K edge XANES spectra, as shown in **Figure R20 (Fig. 4a)**. The AgNO₃ standard had the lowest energy compared to the other samples.

Regarding the question “The impregnation was performed in the same for both Ag/AlOOH, but why the valance differs so much”, our explanation is as follows: In this manuscript, we first obtained γ -Al₂O₃ with different terminal hydroxyl contents by calcining AlOOH at different temperatures, and it is precisely because the terminal hydroxyl group acts as an anchor for Ag on top of the γ -Al₂O₃ that the difference in the terminal hydroxyl content directly resulted in the difference in the dispersion of the Ag species, which in turn led to the difference in the valence state of the Ag species. As a result, the Ag/(Al-900) sample with the highest content of terminal hydroxyl groups shows a monoatomic distribution of Ag exhibiting the oxidized state, whereas the Ag/(Al-500) sample with a low content of terminal hydroxyl groups shows a metallic state with Ag distributed in clusters.

Regarding the question “For the EXAFS data, why there is only Ag-O bond but no Ag-Ag for Ag₂O standard.” We apologize for the omission of the Ag-O-Ag bond

in the EXAFS data of Ag_2O in our original manuscript, which we have corrected in the revised manuscript (**Fig. 4b**).

As you mentioned, “From shape and trend analysis, the peak and FT are similar for $1\text{Ag}/\text{AlOOH-900}$ and Ag_2O .” As we have just explained, it is the difference in the anchoring sites that causes the difference in the dispersion state of the Ag, so the abundance of terminal hydroxyl groups on the $\text{Ag}/(\text{Al-900})$ sample makes the Ag exhibit a single-atom distribution more inclined to the oxidized state, so the peak and FT are similar with Ag_2O . The difference, however, is that there is also a maximum point at Ag-O-Ag in the wavelet transform plot in Ag_2O , whereas the $\text{Ag}/(\text{Al-900})$ sample only has a maximum point at Ag-O, which is the main difference between the wavelet plots of $\text{Ag}/(\text{Al-900})$ and Ag_2O (**Fig. 4d, f**).

In addition to demonstrating the difference in valence between $\text{Ag}/(\text{Al-500})$ and $\text{Ag}/(\text{Al-900})$ samples by XAFS, we also demonstrate the difference in valence by charge density difference, Bader charge data, and density of states (DOS) (**Figure R20 (Fig. 4g-l)**). The substrate transfers an average of 0.176 charge to each Ag atom close to the metallic state in the cluster model ($\text{Ag}/(\text{Al-500})$), whereas Ag transfers 1.036 charge to the substrate, exhibiting an oxidized state, in the monatomic model ($\text{Ag}/(\text{Al-900})$). In addition, the d-band of monatomic Ag has a large band gap at the Fermi energy level and has taken on semiconducting characteristics, suggesting an oxidized state. The d-band of Ag in the cluster model is continuous near the Fermi energy level and takes on a metallic state.

The revision in the manuscript (page 18, line 288-300)

Then we further investigated the charge density difference of Ag species in the $\text{Ag}/(\text{Al-500})$ and $\text{Ag}/(\text{Al-900})$ catalysts. According to the charge density difference and Bader charge data (Fig. 4g-j), it can be seen that the substrate transfers an average of 0.176 charge to each Ag atom in the cluster model ($\text{Ag}/(\text{Al-500})$), whereas Ag transfers 1.036 charge to the substrate in the monatomic model ($\text{Ag}/(\text{Al-900})$), and it is clear that the clusters are close to the metallic state, whereas in the monatomic system, Ag is stabilized by the surrounding OH, and the individual Ag atom

apparently loses its charge and transfers electrons to O, exhibiting an oxidized state. Fig. 4k-l show the density of states (DOS) for the cluster (Ag/(Al-500)) and Ag single atom (Ag/(Al-900)) models. In the cluster model, the d-band of Ag is continuous near the Fermi energy level and is metallic in nature, while the d-band of monatomic Ag has a significant band gap near the Fermi energy level and has taken on semiconducting properties, indicating an oxidized state.

Figure R20 (Fig. 4). Characterization of valence, coordination condition and charge density difference of Ag species.

Reviewer #4 (Remarks to the Author):

The manuscript offered by Li and co-workers outlined a straightforward approach involving carefully regulating the roasting temperature to introduce additional hydroxyl groups to AlOOH, thereby creating anchor sites for Ag species. The authors presented a series of basic characterizations for prepared AlOOH materials including

HR-TEM, HAADF-STEM, FTIR, XAFS, DFT and AIMD simulation. In this way, the existing hypothesis and conclusion cannot be well reconciled. Additionally, the results presented herein may not significantly advance the field beyond the state-of-the-art. The referee considers this issue substantial and critical. More specifically, modification of AlOOH with additional hydroxyl groups is not in itself substantive enough to motivate publication, particularly as the principal technical and scientific conclusions related to the Ag anchoring and its properties are not yet fully developed, consistent, or supported by the theory and the experiments, which is described in more detail below. Consequently, I cannot recommend this work in its current state for publication in Nature Communications.

Response: We sincerely thank you for your valuable comments to help us improve the quality of our manuscript. Based on your and other reviewers' comments, we have done a lot to improve the issues, strengthen the important evidence supporting Ag anchoring, and enhance the connections between the existing hypothesis and the results. The details are as follows: We have supplemented with DFT calculations to demonstrate that the uniformity of the distribution of hydroxyl groups is not the main factor affecting Ag anchoring, and that the number of terminal hydroxyl groups is the key factor determining the dispersion of Ag single atoms. In addition, the physical mechanism of the effect of high-temperature calcination on the formation of hydroxyl groups was elucidated. The difference in surface energy of different crystalline surfaces at varying temperatures determines the transformation of the Al₂O₃ crystalline surface from (110) to (100) under high-temperature calcination, which leads to a significant increase in the content of terminal hydroxyl groups, allowing for monoatomic anchoring of Ag. By complementing ICP and XPS, regarding the issue of whether the distribution of Ag is more even or dense was explained. Additional reactivity and stability tests were supplemented for catalyst applications. Our point-by-point responses are listed below.

#4-1. The author alters the crystal plane of AlOOH upon calcination at varying temperatures, ranging from (110) to (100), with changes corresponding to increasing

temperatures. The verification is solely conducted through HR-TEM, which shows regional limitations (Figure 1). It is crucial to note whether the (100) crystal plane is consistently identified across all locations for AlOOH-900 °C. Additionally, in situ XRD results would be essential for a comprehensive illustration of the transformation in the crystalline structure.

Response: We sincerely appreciate your comment. Based on the “It is crucial to note whether the (100) crystal plane is consistently identified across all locations for AlOOH-900 °C” that you pointed out, we re-characterized the lattice stripes of AlOOH calcined at different temperatures (**Figure R21 (Fig. 1e)**), and additionally provided HR-TEM images of other regions of the Al-500 and Al-900 samples (**Figure R22**). In the newly provided Al-900 HR-TEM image, the 100 facets can be clearly identified in all three regions of the figure. Regarding your question, “The verification is solely conducted through HR-TEM, which shows regional limitations.” In fact, the verification of the facet changes was carried out not only by the HR-TEM but also by AIMD and DFT calculation validation (**Figure R21 (Fig. 1f-i)**), and we verified that the γ -Al₂O₃ (110) facet undergoes a significant structural change that occurs in the γ -Al₂O₃(110) crystal face at 1173 K. The temperature-driven rearrangement of atoms on the surface results in an atomic arrangement that is basically consistent with that of the γ -Al₂O₃ (100) crystal face (**Figure R21 (Fig. 1g-h)**). This suggests that the γ -Al₂O₃ (110) surface has a tendency to change to the γ -Al₂O₃ (100) surface under high temperature calcination conditions. In addition, the surface free energy of γ -Al₂O₃ (110) and γ -Al₂O₃ (100) surfaces at different temperatures (0 K, 300 K, and 1173 K) was calculated by DFT (**Figure R21 (Fig. 1i)**). The calculation results show that the surface energy of γ -Al₂O₃ (110) has a tendency to increase with the increase in calcination temperature, while the surface free energy of γ -Al₂O₃ (100) decreases with the increase in temperature, and the surface energy of γ -Al₂O₃ (100) is lower than that of γ -Al₂O₃ (110) at 1173 K. Therefore, the γ -Al₂O₃ (110) surface structure may be transformed to γ -Al₂O₃ (100) at high temperatures.

Regarding your suggestion “Additionally, in situ XRD results would be essential for a comprehensive illustration of the transformation in the crystalline structure.” We

thank you very much for your valuable comments, and in the revised manuscript, we provide in situ XRD patterns of fresh AlOOH (**Figure R23 (Supplementary Fig. 2)**). From the in-situ XRD results, it can be seen that AlOOH starts to transform into γ -Al₂O₃ at around 400 °C and remains in the γ -Al₂O₃ crystalline phase until the temperature is raised to 900 °C, without any further phase transformation. This result ensures that AlOOH converted to all γ -phase Al₂O₃ at different calcination temperatures excludes the effect of crystalline phase.

The revision in the manuscript (page 6, line 93-100)

The XRD pattern of fresh AlOOH and in situ XRD were used to demonstrate the crystal transition process of AlOOH calcined at different temperatures, as shown in Supplementary Fig. 1-2. The AlOOH crystalline phase persists at calcination temperatures up to 300 °C and starts taking on the crystalline form of γ -Al₂O₃ at 400 °C. Diffraction peaks at 37.5°, 45.7°, 60.5°, 66.6°, and 84.5° were observed, corresponding to the γ -Al₂O₃ (311), (400), (511), (440), and (444) crystal planes (JCPDS 02-1420), which indicates that AlOOH is converted to γ -Al₂O₃ starting at about 400 °C.

Figure R21 (Fig. 1). The crystal plane transformation process of AlOOH calcined at different temperatures, and AIMD and DFT calculations.

Figure R22. Additional supplementary HR-TEM images for Al-500 and Al-900 samples. a, b Al-500. c, d Al-900.

Figure R23 (Supplementary Fig. 2). In situ XRD profiles of AlOOH calcined at different temperatures.

#4-2. Authors have emphasized the function of hydroxyl groups in anchoring Ag atoms. Through calcination, the hydroxyl groups on the surface of the AlOOH are

changed. Specifically, whether there is an increase in the quantity of hydroxyl groups or a more uniform distribution, however, from the results presented here, it is not clear which influence factors are most dominant.

Response: We genuinely thank you for raising this question. According to your suggestion, we have verified that the uniformity of the distribution of hydroxyl groups is not the main factor affecting Ag anchoring by DFT calculations. It is the number of terminal hydroxyl groups that determines the distribution of Ag monoatoms. Specifically, we constructed a model for anchoring Ag atoms with unevenly distributed terminal hydroxyl groups on the (100) surface of γ -Al₂O₃ and uniform bridging hydroxyl groups on the surface of (110), respectively, by means of DFT calculations (**Figure R24 (Fig. 3)**). After structural optimization, we found that on the unevenly distributed γ -Al₂O₃ (100) surface, Ag still showed monoatomic dispersion, while on the uniform (110) surface, Ag was agglomerated. This confirms our conclusion that the number of terminal hydroxyl groups plays a crucial role in regulating the dispersion of Ag species, while the uniformity of the OH distribution plays a secondary role in the anchoring of Ag. We obtained γ -Al₂O₃ with different terminal hydroxyl contents by high temperature calcination to induce the transformation of the crystal surface, and the highest terminal hydroxyl content on the γ -Al₂O₃ (100) surface Al-900 support eventually led to a monatomic distribution of Ag.

The revision in the manuscript (page 14, line 223-233)

In addition to the fact that the number of terminal hydroxyl groups plays an important role in the dispersion of Ag species, it is unclear whether the uniform distribution of hydroxyl groups also has an effect on Ag species. Therefore, we constructed models with unevenly distributed terminal hydroxyl groups on the (100) crystal plane of γ -Al₂O₃ and uniformly distributed hydroxyl groups on the (110) crystal plane to verify this (Fig. 3g). After structural optimization, we found that on the (100) crystal plane, Ag remains in the form of single atoms on the surface of γ -Al₂O₃, while on the (110) crystal plane, Ag agglomerates into small clusters. This indicates that the uniformity

of the hydroxyl group distribution is not the primary factor affecting Ag dispersion; rather, the number of terminal hydroxyl groups is the key factor determining Ag dispersion.

Figure R24 (Fig. 3). DFT calculations

#4-3. In addition to employing the basic DFT and AIMD simulation, the physics regarding the effect of high-temperature calcination on the formation of hydroxyl groups on the AlOOH need to be further clarified. The main reason for this alteration can be attributed to the controlled conditions of calcination, influencing the thermodynamics of hydroxyl group generation and their subsequent distribution on the material surface. The temperature, duration, and atmosphere during calcination play crucial roles in determining whether there is an augmentation in the total number of hydroxyl groups or a more even dispersion of these groups across the surface.

Response: We sincerely appreciate your suggestion. Physically, the surface energy of γ -Al₂O₃ changes during high temperature calcination, which can lead to changes in the surface coordination structure. Our DFT calculations proved that the higher the calcination temperature, the lower the surface energy of the (100) crystal face (**Figure R25 (Fig. 1)**). In addition, our previous study proved that the (100) face has more terminal hydroxyl groups, so by means of high-temperature calcination, it is more favorable for the formation of more terminal hydroxyl groups at 900 °C. According to your suggestion, we performed new experiments to investigate the effect of

temperature, duration, and atmosphere on the hydroxyl content during calcination, as shown in **Figure R26 (Supplementary Fig. 4a-c)**. At a calcination time duration of one hour, the hydroxyl content was highest and almost indistinguishable at 500 °C under N₂, O₂, and air atmospheres, whereas the hydroxyl content was highest and indistinguishable at 900 °C under air and N₂ atmospheres. When the duration was 3 h, the highest hydroxyl content was found in the air atmosphere at 500 °C and 900 °C, and when the duration was increased to 6 h, the highest terminal hydroxyl content was found in the N₂ atmosphere at 500 and 900 °C. Then, we screened the above conditions further (**Figure R26 (Supplementary Fig. 4d)**), and it was found that for 500 °C air atmosphere calcined 3 h and N₂ atmosphere calcined 6 h the hydroxyl content of the difference is not large, while for 900 °C air atmosphere calcined 3 h, the hydroxyl content is much larger than for N₂ atmosphere calcined 6 h.

In summary, the content of terminal hydroxyl groups is highest under the conditions of calcination in air atmosphere for 3 hours. The calcination condition adopted in this manuscript was also calcination in air atmosphere for 3 h. In addition, the issue of the uniformity of hydroxyl group distribution has already been discussed in the previous question #4-2. The number of terminal hydroxyl groups is the most important factor in determining the dispersion of Ag, while the uniformity of hydroxyl group distribution is not the main factor.

The revision in the manuscript (page 11, line 173-181)

In addition, in order to further clarify the physical effects of high-temperature calcination on the formation of hydroxyl groups on AlOOH, we also examined the hydroxyl group changes under different atmospheres and calcination times (Supplementary Fig. 4a-c). Overall, changing the calcination atmosphere and time changed the hydroxyl content to varying degrees, and after screening the conditions, the optimal terminal hydroxyl and total hydroxyl contents were achieved for the 500 °C and 900 °C samples by calcining in an air atmosphere for 3h (Supplementary Fig. 4d), and this calcination condition was also used in this work.

Figure R25 (Fig. 1). Surface free energy of γ -Al₂O₃ (110) and γ -Al₂O₃ (100) structures at different temperatures (0 K, 300 K and 1173 K).

Figure R26 (Supplementary Fig. 4). Hydroxyl changes under different calcination atmospheres and time at 500 and 900 °C calcination temperature.

#4-4. According to the image of 1Ag/AlOOH at different temperatures, the reviewer lacks information regarding whether the distribution of Ag is more even or dense. The XRD results indicate a lack of discernible difference or statistical significance. It is recommended to incorporate additional compelling experiments to strengthen the findings.

Response: We sincerely appreciate your comment. Upon your suggestion, we additionally characterized Ag/(Al-500) and Ag/(Al-900) samples by ICP and XPS to illustrate whether the distribution of Ag was more even or dense. The ICP results (**Figure R27 (Supplementary Fig. 6)**) revealed that the Ag content of both the cluster (Ag/(Al-500)) and monoatomic (Ag/(Al-900)) samples was 0.91 wt%. The XPS results (**Figure R28 (Supplementary Fig. 7)**) demonstrate that the Ag 3d binding energy of the Ag/(Al-900) sample is shifted towards the oxidized state, and the peak intensity is enhanced compared to the Ag/(Al-500) sample, indicating an increase in the number of Ag atoms in the monoatomic sample. Based on the ICP and XPS results, we can conclude that under the condition of maintaining the same Ag content in the cluster and monoatomic samples, the number of Ag atoms in the monoatomic samples increased, indicating that Ag was more equally distributed during the process of converting Ag species from clusters to single atoms. Combined with the HAADF-STEM results, it can be seen more visually that Ag goes from an aggregated cluster state to a monoatomic dispersion for the same Ag content, which can also indicate that the Ag species becomes more evenly dispersed.

The revision in the manuscript (page 23, line 374-379)

During this process, Ag clusters gradually transform into individual Ag atoms, resulting in a more uniform dispersion of Ag species. This phenomenon is confirmed by the results obtained from ICP, XPS, and HAADF-STEM analyses. In both the Ag/(Al-500) and Ag/(Al-900) samples, the Ag content remains consistent. However, the number of Ag atoms on the surface of Ag/(Al-900) samples is notably higher.

Figure R27 (Supplementary Fig. 6). ICP measurement of Ag content of Ag/(Al-500) and Ag/(Al-900) samples.

Figure R28 (Supplementary Fig. 7). XPS Ag 3d spectra of Ag/(Al-500) and Ag/(Al-900) samples.

#4-5. To establish the universality of the material structure, it is suggested to explore alternative systems for verification. It is recommended to examine the performance of alumina-supported silver (Ag/Al₂O₃) catalysts in different catalytic reactions or under varied conditions to provide a more comprehensive assessment of its versatility and applicability.

Response: We genuinely thank you for the constructive comments. According to your suggestion, in order to assess the catalyst's versatility and applicability, the performance and stability of the catalyst were evaluated under different catalytic reactions and reaction conditions. We evaluated the activity of the O₃ decomposition reaction and the stability in the C₃H₆-SCR reaction of Ag/(Al-500) and Ag/(Al-900) samples (**Figure R29 (Fig. 5)**). The ozone conversion of the Ag/(Al-900) sample was maintained at about 55% for 6 h, which is much higher than the Ag/(Al-500) sample of about 15%. The stability results show that within 50 hours, approximately 100% and 75% NO_x conversion rates can be achieved at reaction temperatures of 350 °C and 300 °C, respectively. Additionally, we confirmed the thermal stability of Ag single-atom samples (Ag/(Al-900)) by *ab initio* molecular dynamics simulation (**Figure R29 (Fig. 5d-e)**); during the observation time of 10,000 fs at 723 K, Ag atoms were stably anchored on the surface of γ -Al₂O₃ (100) without agglomeration, suggesting the good thermal stability of Ag/(Al-900).

The revision in the manuscript (page 20-21, line 309-340)

To assess the versatility and applicability of the catalysts, we tested the catalytic activity of the Ag/(Al-500) and Ag/(Al-900) samples in different reactions. HC-SCR technology is considered to be a promising denitrification method, capable of simultaneously removing both NO_x and hydrocarbons from flue gas^{10, 12}. Alumina-supported silver (Ag/Al₂O₃) catalysts find widespread use in HC-SCR applications^{42, 43}. Studies have indicated that highly dispersed silver cations (Ag⁺) are the active centers in this reaction^{11, 15, 44, 45}. As depicted in Fig. 5a, Ag/(Al-X) (X=500, 900) samples were employed to assess the C₃H₆-SCR reactivity, investigating the effects of changes in Ag species dispersion at different calcination temperatures.

Notably, the Ag/(Al-900) samples exhibited higher activity at temperatures ranging from 150 to 400 °C compared to the Ag/(Al-500) samples. We hypothesize that samples calcined at 900 °C with single-atom dispersion are more active than those at 500 °C, which display Ag clusters. Thus, in conjunction with our previous research, it can be further deduced that Ag cations (Ag^+) exhibit enhanced reactivity in the hydrocarbon-selective catalytic reduction of NO_x through HC-SCR. In addition, in the O_3 decomposition reaction (Fig. 5b), the Ag/(Al-900) sample maintained an ozone conversion of about 55% over 6 h, which was much greater than that of the Ag/(Al-500) sample, at about 15%.

A catalyst's stability is another essential indicator of its performance. For the Ag/(Al-900) sample, Fig. 5c displays NO_x conversion of about 100% and 75% for 50 hours at reaction temperatures of 350 °C and 300 °C, respectively. Subsequently, we further verified the stability of Ag/(Al-900) samples through *ab initio* molecular dynamics simulation. As shown in Supplementary Figure 10, the system is stable both in terms of temperature and potential energy. We also calculated the Ag-O and Ag-Ag atomic distances (Fig. 5d), and the results showed that the average distance between Ag and the surrounding O atoms is always shorter than Ag-O bonds within Ag_2O crystals. In addition, the distance between two Ag atoms is always longer than the Ag-Ag bond in the bulk, which indicates that the terminal hydroxyl groups on the $\gamma\text{-Al}_2\text{O}_3$ surface can stably anchor Ag single atoms without agglomeration. Therefore, during the simulation time of 10000 fs, Ag can be stably anchored on the $\gamma\text{-Al}_2\text{O}_3$ (100) surface at 773 K while maintaining the single-atom form (Fig. 5e), suggesting the good thermal stability of Ag/(Al-900).

Figure R29 (Fig. 5). Activity and stability tests.

#4-6. Before such an obvious issue is well resolved, the current manuscript does not allow this referee to endorse it for possible publication on Nat Commun. The manuscript requires a major revision. Please see some minor comments below to improve the manuscript:

1) Introduction was poorly drafted: it has both focus issue and logic issue. Re-writing is necessary.

Response: We thank you for the important comment. In terms of writing focus, we replaced the original focus on the preparation of single-atom catalysts with a focus on hydroxyl group anchoring. The changes in writing logic are as follows: from the previous studies proposing the “terminal hydroxyl group anchoring mechanism” to the recent studies supporting the universality of this anchoring mechanism on other metals and supports. We found that the type of crystal facets exposed by the metal oxides (Al_2O_3 (100), CeO_2 (100)) significantly affects the content of terminal hydroxyl groups. However, it is unclear how to regulate the content of terminal hydroxyl groups on the surface of the support, as well as the type of exposed crystalline surfaces, and whether the transformation of the type of crystalline surfaces impacts the type and content of hydroxyl groups. According to the fact that the surface energy of crystalline facets changes at different temperatures, in this work, Ag/Al catalysts with different hydroxyl content were prepared by inducing facet transformation through high-temperature calcination, and modulation of the terminal hydroxyl group was achieved and thermally stable Ag single-atom catalysts successfully prepared. Based on the above focus and logic, the re-written introduction has been corrected in the revised manuscript.

The revision in the manuscript (page 3-5, line 33-81)

In heterogeneous catalysis, a strong metal-support interaction (SMSI) is deemed to be a key factor affecting the anchoring of metals on the surface of reducible oxides and plays a significant role in influencing the dispersion of the active component^{1,2}, the electron migration between the metal and the support, and thus the performance of the catalytic reaction. However, there are almost no reports regarding SMSI on non-reducible oxide supports. Nevertheless, catalysts supported on non-reducible oxides are also widely applied. For non-reducible oxides like Al_2O_3 , which serves as a support for the widely used Ag/ Al_2O_3 catalysts, typical reactions include selective catalytic oxidation of NH_3 ($\text{NH}_3\text{-SCO}$)^{3,4}, selective catalytic reduction (SCR) of NO_x with NH_3 ($\text{NH}_3\text{-SCR}$)⁵⁻⁷, CO oxidation^{8,9}, (hydrocarbon-based selective catalytic reduction of NO_x) HC-SCR¹⁰⁻¹², ethylene epoxidation reaction^{13,14}, etc. Therefore, it

is necessary to clarify the anchoring mechanism of Ag on the Al₂O₃ surface to regulate the state of Ag active species and thus enhance the catalytic reaction performance.

In our previous study¹⁵⁻¹⁷, we used two types of γ -Al₂O₃, with different hydroxyl contents, nano-sized γ -Al₂O₃ and micro-sized γ -Al₂O₃, and found that Ag species are mainly anchored to γ -Al₂O₃ through terminal hydroxyl group. DFT calculations revealed that the (100) surfaces of γ -Al₂O₃ have more terminal hydroxyl groups than the (110) surfaces, allowing Ag to be anchored as a single atom by the terminal hydroxyls. When not enough terminal hydroxyl groups are available to anchor the Ag species, they tend to aggregate into Ag clusters and AgNPs¹⁵. Based on the proposed “terminal hydroxyl group anchoring mechanism,” we constructed more effective NH₃-SCO active sites using the “pre-occupied anchoring-site” strategy¹⁶. We used Cu with stronger anchoring strength to pre-occupy the anchoring sites to force Ag agglomeration, resulting in the construction of an efficient NH₃-SCO catalyst at low Ag loading. Meanwhile, we found that the “terminal hydroxyl group anchoring mechanism” is also applicable to the anchoring of other non-precious metals (Fe, Co, Ni, and Mn) on Al₂O₃ support. In addition, we found that on the CeO₂ support, the terminal hydroxyl group is also an anchoring site for Ag atoms, and terminal OH group on the CeO₂ (100) surface can firmly anchor Ag via the formation of a dumbbell structure¹⁷. Moreover, other metals (Pt, Pd, etc.) on CeO₂ can also be directly anchored to terminal hydroxyl group.

As mentioned above, the above findings suggest that terminal hydroxyl groups on oxide surfaces are the anchoring sites for metals, and that the type of exposed crystal facets (Al₂O₃ (100), CeO₂ (100)) on the metal oxide significantly affects the content of the terminal hydroxyl groups, and thus the dispersion of the metal. However, it is not clear how to regulate the content of terminal hydroxyl groups on the support surface or the type of exposed crystal faces and whether the transformation of the crystal plane type correspondingly affects the type and content of hydroxyl groups. This is thus clearly worthy of further investigation. In conventional nanocatalysis, the influence of the crystal facet effect of supports on the

activity of the catalytic reaction has been extensively studied. Hu et al¹⁸. loaded Pd on different crystal facets of CeO₂ and found that on the CeO₂ (100) facets, Pd exists predominantly in the form of Pd SAs (single atoms). In contrast, on the CeO₂ (111) facets, Pd readily aggregates into Pd clusters. Notably, when the size of the catalytically active components goes from the nanoscale to the single-atom scale, the crystal facet effect becomes more noticeable and must be considered.

Physically, when a crystal surface is exposed to different thermodynamic conditions, a change in surface energy occurs, which can lead to the possibility of atomic rearrangement.....

2) The abscissa of Fig.3d shown “wavenumbers” is totally wrong.

Response: We sincerely appreciate you pointing out the mistakes. We’ve corrected this mistake in the revised manuscript (Figure R30 (Fig. 2d)).

Figure R30 (Fig. 2). Hydroxyl change characterization and Ag anchoring mechanism schematic diagram.

3) The corresponding crystal plane is not marked in Figure 1c, d.

Response: Thank you for pointing this out. We have added the missing crystal planes in the revised manuscript (Figure R31 (Fig. 1c, d)).

Figure R31 (Fig. 1). The crystal plane transformation process of AlOOH calcined at different temperatures, and AIMD and DFT calculations.

4) There is no definition or standard abbreviation for the position where “HC-SCR” appears for the first time in the manuscript.

Response: We sincerely thank you for the kind reminder. We have defined where HC-SCR first appears in the revised manuscript.

The revision in the manuscript (page 3, line 42-43)

(hydrocarbon-based selective catalytic reduction of NO_x) HC-SCR

REVIEWERS' COMMENTS

Reviewer #1 (Remarks to the Author):

The authors have answered the reviewer's comments well and I recommend publication of this article.

Reviewer #2 (Remarks to the Author):

The revised manuscript has solved the issue I raised and in my opinion it also properly answered the problems raised by the other reviewers. Therefore I believe this revised manuscript can be published in Nature Communications.

Reviewer #3 (Remarks to the Author):

My technical questions were addressed. I remain neutral on the novelty part, as I still feel that the discovery is incremental, rather than substantial, in light of the fact that their previous publication at Nat. Commun. 2020, 11(1), 529 has already covered the key points.

Reviewer #4 (Remarks to the Author):

All of comments have been addressed accordingly. The paper can be published as is.